eLIFE elifesciences.org

# Mutant huntingtin enhances activation of dendritic Kv4 $K^+$ channels in striatal spiny projection neurons

**Luis Carrillo-Reid[1,2], Michelle Day[1], Zhong Xie[1], Alexandria E Melendez[1], Jyothisri Kondapalli[1], Joshua L Plotkin[1,3], David L Wokosin[1], Yu Chen[1], Geraldine J Kress[1,4], Michael Kaplitt[5], Ema Ilijic[1], Jaime N Guzman[1], C Savio Chan[1], D James Surmeier[1]***

[1]Department of Physiology, Feinberg School of Medicine, Northwestern University, Chicago, United States; [2]Department of Developmental Neurobiology and Neurophysiology, Neurobiology Institute, National Autonomous University of Mexico, Queretaro, Mexico; [3]Department of Neurobiology & Behavior, Stony Brook University School of Medicine, Stony Brook, United States; [4]Department of Neurology, Washington University School of Medicine, St. Louis, United States; [5]Department of Neurological Surgery, Weill Cornell Medical College, New York, United States

**Abstract** Huntington's disease (HD) is initially characterized by an inability to suppress unwanted movements, a deficit attributable to impaired synaptic activation of striatal indirect pathway spiny projection neurons (iSPNs). To better understand the mechanisms underlying this deficit, striatal neurons in ex vivo brain slices from mouse genetic models of HD were studied using electrophysiological, optical and biochemical approaches. Distal dendrites of iSPNs from symptomatic HD mice were hypoexcitable, a change that was attributable to increased association of dendritic Kv4 potassium channels with auxiliary KChIP subunits. This association was negatively modulated by TrkB receptor signaling. Dendritic excitability of HD iSPNs was rescued by knocking-down expression of Kv4 channels, by disrupting KChIP binding, by restoring TrkB receptor signaling or by lowering mutant-Htt (mHtt) levels with a zinc finger protein. Collectively, these studies demonstrate that mHtt induces reversible alterations in the dendritic excitability of iSPNs that could contribute to the motor symptoms of HD.
DOI: https://doi.org/10.7554/eLife.40818.001

***For correspondence:** j-surmeier@northwestern.edu

**Competing interests:** The authors declare that no competing interests exist.






## Introduction

HD is an autosomal dominant neurodegenerative disorder caused by an expanded polyglutamine repeat in the huntingtin gene (***Zuccato et al., 2010***). At the outset of motor symptoms, HD patients have uncontrolled, choreic 'dance-like' movements. These symptoms are accompanied by down-regulation of phenotypic markers in striatal indirect pathway spiny projection neurons (iSPNs), which has led to the hypothesis that the hyperkinetic symptoms are due to a loss in indirect pathway function (***Albin et al., 1992***).

What causes the deficit in iSPNs is less clear. Study of genetic HD models has implicated a large cast of potential factors that could be responsible. One of these factors is the loss of neurotrophic support (***Gil and Rego, 2008***). Mutant Htt (mHtt) induces deficits in the synthesis, transport and release of brain derived neurotrophic factor (BDNF) (***Zuccato et al., 2010***), as well as deficits in post-synaptic BDNF signaling through TrkB receptors (TrkBRs) (***Ginés et al., 2010***; ***Plotkin et al., 2014***).

In mouse models of HD, the progressive loss of TrkBR signaling leads to the disruption of long-term potentiation (LTP) at iSPN corticostriatal glutamatergic synapses (**Plotkin et al., 2014**). As corticostriatal glutamatergic synapses are the principal source of excitatory input to iSPNs (**Bolam et al., 2000**), the hyperkinetic symptoms of early stage HD could be due, in part, to a deficit in the ability to appropriately activate iSPNs (**Plotkin and Surmeier, 2015**).

In addition to a deficit in synaptic plasticity, diminished dendritic excitability also could contribute to the failure of iSPNs to be activated by cortical activity. Kv4 $K^+$ channels are key determinants of iSPN dendritic excitability; these channels dampen dendritic propagation of excitatory postsynaptic potentials (EPSPs), as well as somatically generated back-propagating action potentials (bAPs) (**Day et al., 2008**). Befitting their importance in dendritic integration, Kv4 $K^+$ channels are modulated by several signaling cascades. For example, extracellular signal-regulated protein kinases (ERKs), which are potently activated by TrkBRs, phosphorylate Kv4 $K^+$ channels limiting their activation in response to membrane depolarization (**Hu et al., 2006**; **Schrader et al., 2009**; **Yang et al., 2001**). Kv4 $K^+$ channels also are modulated by local $Ca^{2+}$ signaling; $Ca^{2+}$ influx through Cav3 $Ca^{2+}$ channels has the opposite effect of TrkBR signaling, diminishing inactivation of Kv4 channels and increasing their ability to blunt sub-threshold depolarizing events (**Anderson et al., 2010**). This modulation requires an auxiliary subunit referred to as a $K^+$ channel interacting protein (KChIP) (**Anderson et al., 2010**; **Shibata et al., 2003**; **Vacher and Trimmer, 2011**). Interestingly, although it is unclear why, KChIPs are also required for ERK phosphorylation to negatively modulate Kv4 channel gating (**Schrader et al., 2006**).

All three of these elements – KChIPs, Cav3 $Ca^{2+}$ channels and TrkBRs – richly invest the distal dendrites of SPNs (**Carter and Sabatini, 2004**; **Lobo et al., 2010**; **Plotkin et al., 2011**; **Rhodes et al., 2004**) raising the possibility that they bidirectional regulate Kv4 $K^+$ channel gating, as described in other neurons. This is relevant to HD as the loss of TrkBR signaling in iSPNs could enhance Kv4 $K^+$ channel availability and diminish the responsiveness of iSPNs to excitatory synaptic signals necessary for movement suppression. This study was undertaken to test this hypothesis. Indeed, the dendritic excitability of iSPNs in two genetic mouse models of HD (BACHD and Q175) was depressed at an age (~6 months old) when mice became symptomatic. The reduction in dendritic excitability could be traced to enhanced opening of Kv4 $K^+$ channels arising from increased association of Kv4 channels with auxiliary KChIP subunits and Cav3 $Ca^{2+}$ channels. This association was negatively modulated by TrkBR signaling in wild-type mice but not in HD mice. Pharmacological or genetic suppression of p75NTR signaling pathways restored the ability of TrkBR signaling to modulate Kv4 channels in HD iSPNs. Finally, striatal suppression of mHtt transcription with a zinc finger protein (ZFP) (**Garriga-Canut et al., 2012**) reversed the iSPN deficits in both dendritic excitability and synaptic plasticity pointing to a regionally-autonomous mechanism in the disease phenotype.

# Results

Two genetic models of HD were used in these experiments: hemizygous BACHD mice (**André et al., 2011**; **Gray et al., 2008**) and heterozygous Q175 knock-in mice (Q175+/-) (**Heikkinen et al., 2012**; **Menalled et al., 2012**). The BACHD mouse is a transgenic model of HD in which the full-length human mHtt has been inserted using a bacterial artificial chromosome (BAC). These mice display progressive motor and physiological deficits. At 2–3 months of age, the deficits are very modest but become more discernible by 6 months of age. To cross validate results in the BACHD model, Q175 ±mice having an Htt gene with an expanded CAG locus (175–180 repeats) were also examined; these mice also manifest a progressive motor and physiological phenotype that becomes apparent at 6 months of age (**Heikkinen et al., 2012**; **Menalled et al., 2012**). To selectively sample iSPNs and neighboring direct pathway SPNs (dSPNs), these HD models were crossed into reporter lines for these two cell types (**Gerfen and Surmeier, 2011**).

## Dendritic excitability was reduced in iSPNs from HD models

To characterize the excitability of SPN dendrites, patch clamp and optical approaches were used in ex vivo striatal brain slices (**Carter et al., 2007**; **Day et al., 2008**; **Higley and Sabatini, 2010**; **Plotkin et al., 2011**). To visualize dendritic trees, cells were loaded with red Alexa Fluor 568 (25 µM) with a somatic patch electrode (**Figure 1a**). Previous work has shown that action potentials (APs) evoked in the soma of SPNs by brief current steps, back-propagate into SPN dendrites. These back-

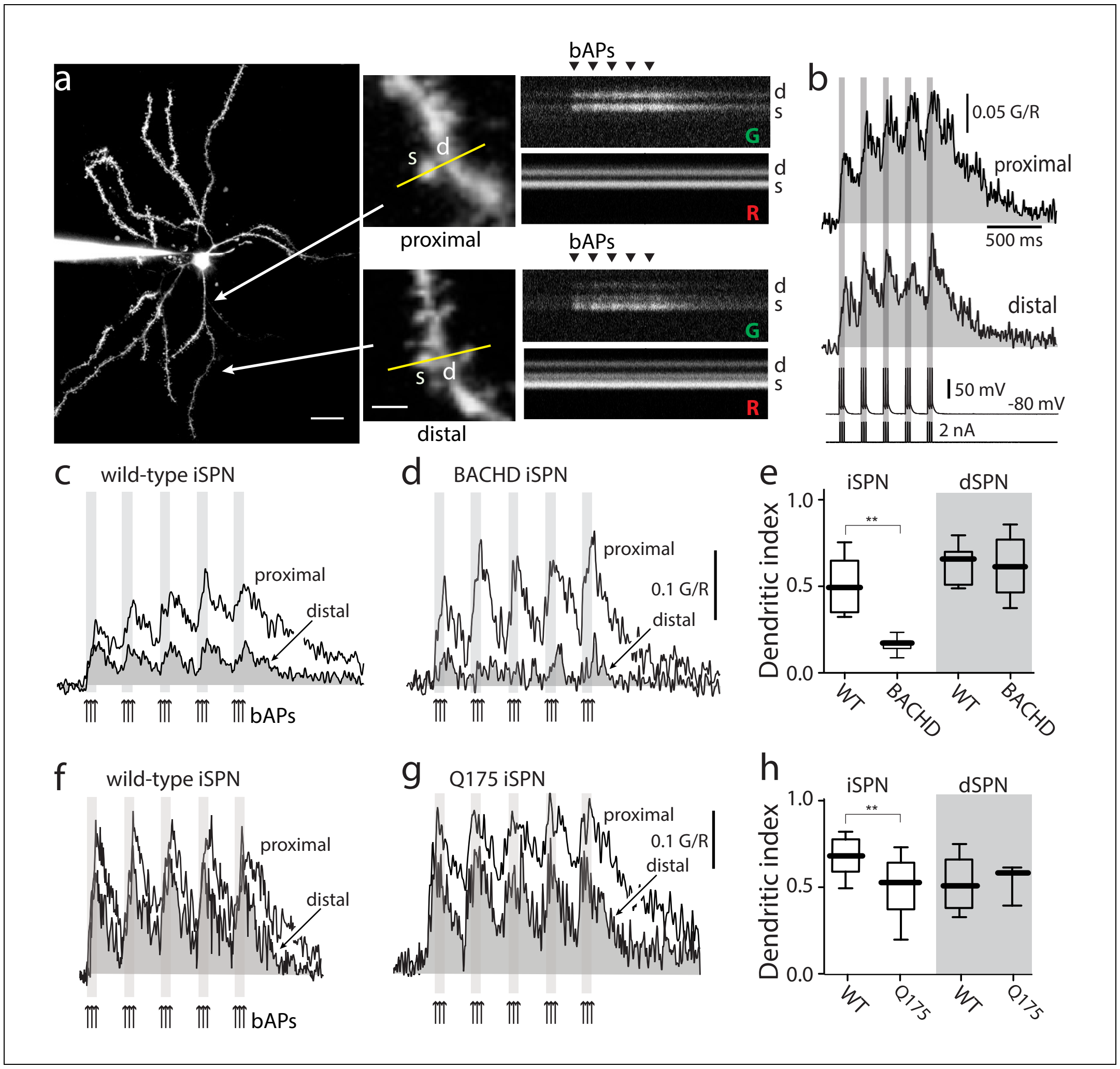


**Figure 1.** Distal dendritic excitability was reduced in iSPNs from HD models. (**a**) Characterization of $Ca^{2+}$ transients induced by bAPs in dendritic spines of SPNs. Left: Maximum intensity Z projection images of an iSPN filled with Alexa Fluor 568 (wild-type mouse >2 months; scale bar 20 μm). Middle: Line scans taken in proximal (~50 μm) and distal (~100 μm) spines (letter s) and dendrites (letter d) located in the same focal plane (scale bar 2 μm). Right: Calcium transients evoked by bAPs (triangles). (**b**) Quantification of the area under the curve from $Ca^{2+}$ transients taken at proximal (first row) and distal (second row) head spines; similar transients were seen in dendritic shafts (***Figure 1—figure supplement 1***). Voltage responses (third row; 5 burst of 3 APs each) evoked by somatic current injection (fourth row; 2 nA amplitude; 2 ms duration; inter burst frequency: 5 Hz; intra burst frequency: 50 Hz). (**c**) $Ca^{2+}$ transients evoked by bAPs (arrows) at proximal and distal dendritic sites in iSPNs from wild-type (WT) mice and (**d**) symptomatic BACHD mice. Here, the dendritic $Ca^{2+}$ signal was derived from shafts; similar transients were seen in spines. (**e**) Dendritic index representing the ratio of the area under the curve between distal and proximal $Ca^{2+}$ transients taken from dendritic spines and shafts in iSPNs and dSPNs demonstrates the reduction of dendritic excitability in iSPNs taken from symptomatic BACHD mice (left) (p=0.0008, Mann-Whitney U, Two-Tailed, n = 9). Dendritic index was not significantly different in dSPNs (right) from WT and BACHD mice (p=0.95216, Mann-Whitney U, Two-Tailed, n = 8). (**f, g**) $Ca^{2+}$ transients from Q175 ±iSPNs and their WT littermates. (**h**) The boxplot shows that the dendritic index was reduced in Q175 ±iSPNs compared to WT littermates (left; p=0.0107, Mann-Whitney U, One-Tailed, n = 6). Dendritic index was not significantly different in dSPNs WT and Q175 ±mice (right; p=0.15; Mann-Whitney U, Two-tailed, n = 5 WT, n = 3 Q175+/-). See ***Figure 1—source data 1***.

DOI: https://doi.org/10.7554/eLife.40818.002

The following source data and figure supplements are available for figure 1:

**Source data 1.** Source data for ***Figure 1***.

DOI: https://doi.org/10.7554/eLife.40818.007

*Figure 1 continued*

**Figure supplement 1.** bAP-evoked distal but not proximal dendritic fluorescence measurements were diminished in BACHD iSPNs.

DOI: https://doi.org/10.7554/eLife.40818.003

**Figure supplement 1—source data 1.** Source data for ***Figure 1—figure supplement 1***.

DOI: https://doi.org/10.7554/eLife.40818.004

**Figure supplement 2.** Expression of Cav3 mRNAs was not significantly altered in HD iSPNs.

DOI: https://doi.org/10.7554/eLife.40818.005

**Figure supplement 2—source data 1.** Source data for ***Figure 1—figure supplement 2***.

DOI: https://doi.org/10.7554/eLife.40818.006

propagating APs (bAPs) transiently open voltage-dependent $Ca^{2+}$ channels, which elevates cytosolic $Ca^{2+}$ concentration (***Carter et al., 2007***; ***Day et al., 2006***). The bAP-evoked $Ca^{2+}$ transient can be used as a surrogate measure of dendritic excitability as SPN dendrites are too small to directly record from with a patch electrode. To measure this transient, neurons were dialyzed with the $Ca^{2+}$ dye Fluo4 (100 µM) and dendritic fluorescence measured using two photon laser scanning microscopy (2PLSM). Brief trains of somatic bAPs were generated and line scans performed at parfocal proximal (~50 µm from soma) and distal (~100 µm from soma) dendritic locations (***Figure 1b***).

The initial evaluation of the bAP-evoked fluorescence signals revealed no change in proximal dendritic fluorescence of wild-type and BACHD iSPNs but a drop in that of the distal dendrites of BACHD iSPNs (***Figure 1—figure supplement 1***). To generate reliable quantitative $Ca^{2+}$ estimates from fluorescence measurements three steps were taken. First, the area of the fluorescence response above baseline was calculated to generate a measure of the net $Ca^{2+}$ charge transferred; this minimized variability associated with dye concentrations and buffering. Second, a red, $Ca^{2+}$-insensitive dye was used to locate dendrites and to monitor dye loading and changes in laser power or focal plane. Third, the $Ca^{2+}$ area estimate in distal dendrites was normalized by that in a proximal parfocal dendrite. This ratio will be referred to as the dendritic index. In iSPNs from 6 month old BACHD mice, the dendritic index of iSPNs was significantly smaller than that from wild-type iSPNs indicating a relative reduction in distal dendritic excitability (***Figure 1c–e***). In contrast, there was no difference in the dendritic index of age-matched dSPNs from BACHD mice (***Figure 1e***). A qualitatively similar loss of distal dendritic excitability was found in iSPNs from 6 month old Q175 ±mice (***Figure 1f–h***). As for BACHD dSPNs, Q175 ±dSPNs did not differ significantly from their wild-type littermates. Because all of the phenomena observed in iSPNs were the same in the two mouse models, for simplicity the two will be lumped and called HD mice (genotype data can be found in the figure legends).

Previous work had shown that there was no discernible change in the dendritic architecture of HD iSPNs at this age (***Indersmitten et al., 2015***; ***Plotkin et al., 2014***), so physiological factors that might account for the drop in dendritic index were explored. One potential explanation for this change is that Cav3 $Ca^{2+}$ channels, which are preferentially located in distal dendrites of SPNs (***Plotkin et al., 2011***), were down-regulated in HD iSPNs. However, quantitative polymerase chain reaction (qPCR) analysis of mRNAs in iSPNs separated by fluorescence activated cell sorting (FACS) didn't reveal any significant change in the expression of either of the two primary Cav3 channel pore-forming subunits (Cav3.2, Cav3.3) (***Figure 1—figure supplement 2***).

## Reduced dendritic excitability was dependent upon Kv4 channels

Another determinant of bAP propagation and dendritic excitability in SPNs is the voltage-dependent Kv4 $K^+$ channel (***Day et al., 2008***). The attenuation of bAP-evoked dendritic $Ca^{2+}$ transients in HD iSPNs could be a consequence of increased opening of these channels (***Johnston et al., 2003***). To test this hypothesis, the striatum of wild-type and HD mice were injected with adeno-associated virus (AAV) containing a bicistronic expression cassette for a Kv4.2 shRNA and enhanced green fluorescent protein (eGFP). Four weeks after AAV injection, Kv4.2 mRNA was significantly reduced in the striatum (***Figure 2—figure supplement 1a–b***). Moreover, in infected iSPNs (***Figure 2a***), the relationship between somatically injected current and the frequency of spiking in HD iSPNs was shifted in a way expected from Kv4 knockdown (***Figure 2—figure supplement 1c–d***). More importantly iSPN dendrites also were more excitable as bAPs evoked a robust $Ca^{2+}$ transient in both proximal and

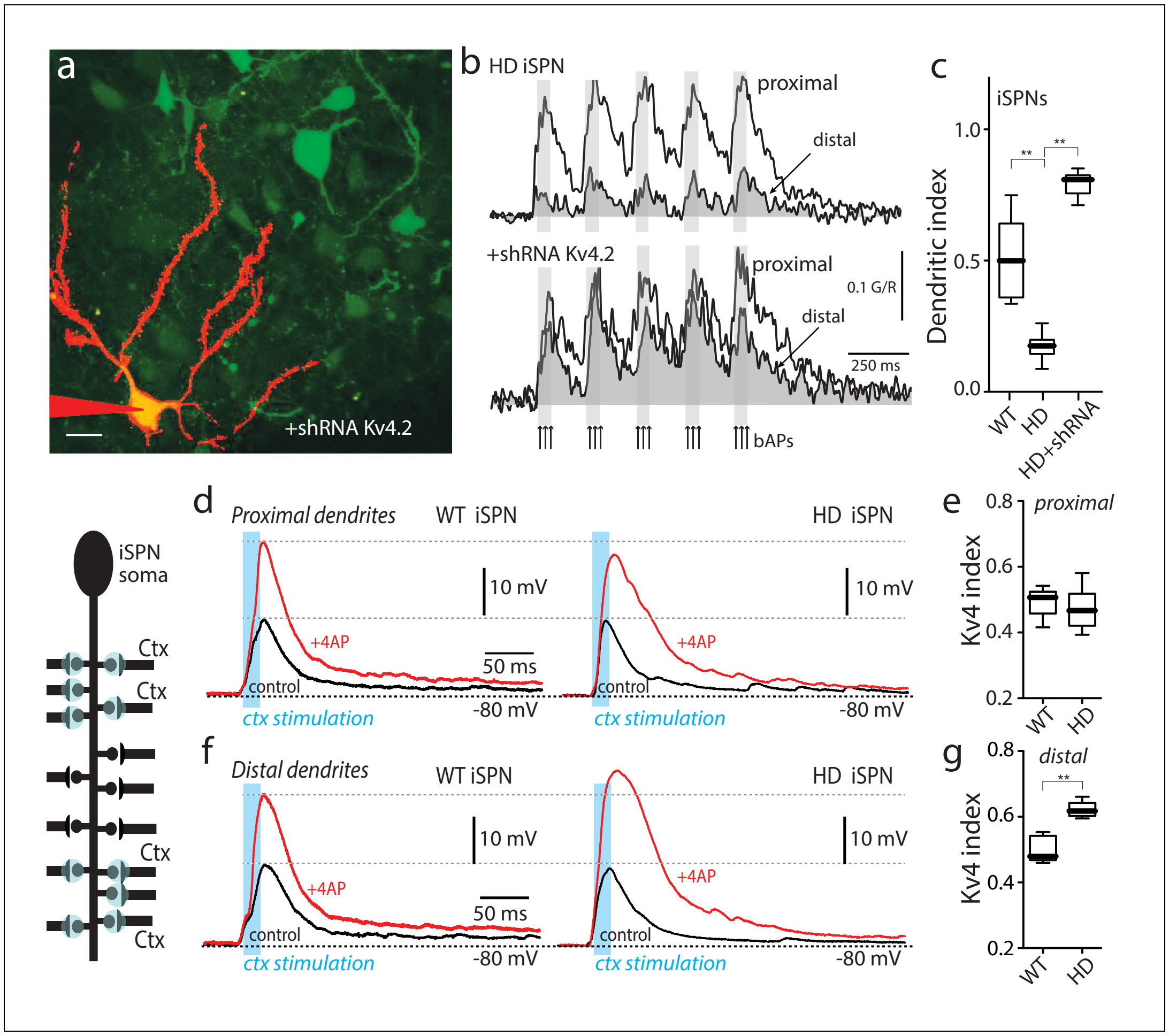


**Figure 2.** Upregulation of Kv4 channel activity suppressed dendritic excitability in HD iSPNs. (**a**) Viral delivery of Kv4.2 shRNA into the striatum of BACHD mice. Z projection images of an iSPN filled with Alexa Fluor 568 and transfected with Kv4.2 shRNA for 4 weeks. Infected neurons are labeled by eGFP, note the presence of the label in the soma and along the dendrites of SPNs (scale bar 10 μm). (**b**) Calcium transients in response to bAPs (arrows) at proximal and distal dendritic spines in iSPNs from BACHD mice; the loss of dendritic excitability observed in non-transfected neurons (top) was rescued by knocking down Kv4.2 channels (bottom). (**c**) The dendritic index was significantly smaller in non-infected neurons than in infected iSPNs (p=0.0002, Mann-Whitney U, Two-Tailed, n = 6). (**d**) Voltage responses to optogenetic stimulation of corticostriatal terminals at proximal dendrites in WT (left) and HD (right) iSPNs before and after application of 4-AP (2 mM). (**e**) Kv4 index reflecting the engagement of Kv4.2 potassium channels is not significantly different in proximal dendrites of WT compared with BACHD mice (p=0.3401, Mann-Whitney U, Two-Tailed, n = 9). (**f**) Voltage responses to optogenetic stimulation of corticostriatal terminals at distal dendrites in WT (left) and BACHD (right) iSPNs before and after 4-AP (2 mM) application. (**g**) Kv4 index was larger in distal dendrites of HD iSPNs than in WT iSPNs (p=0.0007, Mann-Whitney U, Two-Tailed, n = 9). See ***Figure 2—source data 1***.

DOI: https://doi.org/10.7554/eLife.40818.008

The following source data and figure supplements are available for figure 2:

**Source data 1.** Source data for ***Figure 2***.

DOI: https://doi.org/10.7554/eLife.40818.014

**Figure supplement 1.** AAV Kv4.2-shRNA mediated knockdown in striatum increased iSPN excitability.

DOI: https://doi.org/10.7554/eLife.40818.009

**Figure supplement 1—source data 1.** Source data for ***Figure 2—figure supplement 1***.

DOI: https://doi.org/10.7554/eLife.40818.010

**Figure supplement 2.** 4-AP (2 mM) increased EPSP amplitude only when 5–10 spines were stimulated.

DOI: https://doi.org/10.7554/eLife.40818.011

**Figure supplement 2—source data 1.** Source data for ***Figure 2—figure supplement 2***.

*Figure 2 continued*

DOI: https://doi.org/10.7554/eLife.40818.012

**Figure supplement 3.** Computation of the Kv4-index.

DOI: https://doi.org/10.7554/eLife.40818.013

distal dendrites (***Figure 2b–c***). In fact, the relative excitability of distal dendrites of HD iSPNs after Kv4.2 knockdown was significantly greater than that of wild-type iSPNs, in agreement with previous studies implicating Kv4 $K^+$ channels in the regulation of basal excitability of distal dendrites (***Day et al., 2008***).

One caveat of using bAPs to probe dendritic excitability is that alterations in the ability of proximal dendrites to propagate activity could be misinterpreted as a change in distal excitability. To directly interrogate distal dendrites, optogenetic approaches were used. For these experiments a mouse line of transgenic mice, in which channelrhodopsin2 (ChR2) was expressed in a large percentage of cortical pyramidal neurons (***Arenkiel et al., 2007***), was crossed into the BACHD iSPN reporter line. In the presence of tetrodotoxin (1 μM) to block conducted activity and 4-aminopyridine (4-AP, 200 μM) to enhance presynaptic excitability (***Petreanu et al., 2009***), ChR2-containing presynaptic terminals abutting spine heads were activated by a blue (473 nm) laser flash (spot size ~10 μm in diameter). Laser intensity and the number of spine heads stimulated in rapid succession were adjusted to yield a somatic excitatory postsynaptic potential (EPSP) of about 10 mV. Based upon previous work, this should have arisen from 5 to 10 neighboring synapses (***Plotkin et al., 2014***).

To assess the role of Kv4 $K^+$ channels in regulating EPSPs, the extracellular concentration of 4-AP was increased to 2 mM after measuring the basal EPSP. Given the differences in the affinity of 4-AP for Kv1 and Kv4 channels (***Hille, 2001***), this should allow the contribution of Kv4 channels to be determined. At synapses on proximal iSPN dendrites, increasing the 4-AP concentration roughly doubled the response to optogenetic activation of corticostriatal terminals (***Figure 2d***, left). It is worth noting that 2 mM 4-AP had little effect on EPSPs when they were in the 0.5–1 mV range at the soma (***Figure 2—figure supplement 2***), suggesting that Kv4 channels were largely postsynaptic and required a significant depolarization to be activated. The magnitude of the 4-AP induced enhancement of EPSPs was similar at proximal synapses on HD and wild-type iSPNs (***Figure 2d***, right). To quantify the enhancement, a 'Kv4 index' was defined (1 R, where R was the ratio of EPSP amplitudes before and after 2 mM 4-AP; this index grows larger with increasing engagement of Kv4 channels; ***Figure 2—figure supplement 3***). As inferred from the previous observations, the Kv4 index at proximal synapses was not different between wild-type and HD iSPNs (***Figure 2e***). At distal synapses, 2 mM 4-AP also roughly doubled the amplitude of the evoked EPSP in wild-type iSPNs (as in proximal dendrites), but in HD iSPNs the Kv4 index was significantly larger (***Figure 2f,g***). This difference was consistent with the proposition that in HD iSPNs, distal dendrites were less excitable because Kv4 $K^+$ channels were more robustly engaged by depolarization.

## Upregulation of Kv4 channels was dependent upon KChIPs

Functional Kv4 channels are macromolecular complexes (***An et al., 2000***; ***Birnbaum et al., 2004***; ***Shibata et al., 2003***). Quantitative PCR analysis of iSPNs separated by FACS revealed that, in addition to mRNAs for Kv4.1, Kv4.2, Kv4.3 subunits, iSPNs expressed mRNAs for auxiliary KChIP subunits (***Figure 3—figure supplement 1***). However, there were no significant differences in Kv4 or KChIP mRNA expression between iSPNs taken from wild-type and HD mice. The qPCR results also make the point that mRNA for Kv4.2 subunits was much more abundant in iSPNs than mRNAs for either Kv4.1 or Kv4.3 subunits.

Although Kv4 mRNA expression was unchanged, it could be the case that the subcellular distribution of channels shifted toward distal dendrites in Q175 iSPNs, resulting in distal hypoexcitability. To test this possibility, iSPNs were sparsely labeled by injecting a low titre of AAV carrying a Cre-dependent tdTomato expression construct into the striatum of 6 month-old wildtype and Q175 A2a-Cre mice. After allowing time for expression of the construct, mice were sacrificed, perfused and processed for Kv4.2 immunoreactivity (IR). Next, the density of Kv4.2 IR in proximal (<80 μm from soma) and distal (>100 μm from soma) dendrites of iSPNs was assessed from deconvolved, confocal z-stacks using a proximity algorithm (Imaris Coloc, see Materials and methods). This analysis failed to

find any difference in the density of Kv4.2 IR in proximal and distal dendrites of wildtype and Q175 iSPNs (***Figure 3—figure supplement 2***).

An alternative hypothesis is that an increased association between KChIP and Kv4 subunit proteins was responsible for elevated channel function in HD iSPNs. To test this hypothesis, iSPNs in ex vivo brain slices were dialyzed with an antibody that disrupts the KChIP interaction with the Kv4 channel (***Anderson et al., 2010***). Dialysis with a sub-type independent KChIP antibody (pan-KChIP; 1:50) rescued dendritic excitability in HD iSPNs (***Figure 3—figure supplement 3***). Because KChIP2 is striatally enriched (***Rhodes et al., 2004***; ***Xiong et al., 2004***), a subtype-specific antibody was tested. Dialysis of the KChIP2 antibody also rescued dendritic excitability in HD iSPNs, whether measured using a bAP protocol or synaptic stimulation (***Figure 3a–d***).

KChIPs have been shown to promote Kv4 channel function in two ways (***Vacher and Trimmer, 2011***). One way is to increase Kv4 protein expression and surface trafficking. However, Kv4.2 protein in the HD striatum was not greater than that in littermate controls (***Figure 3e,f***) suggesting this was not their mode of action in HD iSPNs (although a relative increase in surface expression cannot be excluded). Another way in which KChIP2 enhances Kv4 channel function is by increasing channel availability in response to depolarization and an elevation in intracellular $Ca^{2+}$ concentration. Cav3 $Ca^{2+}$ channels, which are enriched in distal dendrites of SPNs (***Plotkin et al., 2011***), can form a signaling complex with KChIPs and Kv4 channels in cerebellar neurons (***Anderson et al., 2010***). Indeed, Kv4.2 and Cav3.2 subunits could be co-immunoprecipitated from striatal tissue (***Figure 3g***) suggesting that they form a similar complex in iSPNs. This association was specific to Cav3 channels, as Cav1.3 $Ca^{2+}$ channels did not co-immunoprecipitate with Kv4.2 channels (***Figure 3—figure supplement 4***). Moreover, consistent with previous work (***Anderson et al., 2010***), the preferential Cav3 channel inhibitor mibefradil (3 μM) blunted the Kv4 channel attenuation of distal, optogenetically evoked EPSPs in HD iSPNs (***Figure 3h,i***).

## TrkBR signaling regulated Kv4 channel association with KChIPs

To determine whether there was a physical association between Kv4.2 and KChIP proteins, the ability of a Kv4.2 subunit antibody to pull down KChIP proteins from striatal homogenates was examined. Both KChIP1 and KChIP2, but not KChIP3/4, were robustly pulled down with a Kv4.2 antibody (***Figure 4a***). The banding pattern of KChIP isoforms was similar to that in previous reports (***An et al., 2000***), suggesting that a common set of splice variants was present. To provide an additional test of the association between KChIPs and Kv4.2, the ability of a pan-KChIP antibody to pull down Kv4.2 subunits was examined. As expected, the pan-KChIP antibody was able to pull down Kv4.2 protein (***Figure 4—figure supplement 1a***). However, antibody to other potassium channel subunits such as Kv1.2 was unable to pulldown KChIPs (***Figure 4—figure supplement 1b***), and non-immune IgG did not pull down either Kv4.2 (not shown) or Cav3.2 protein (***Figure 3g***), supporting the notion that Kv4.2-KChIP interaction was specific. In striatum, KChIP2 expression is enriched in SPNs and KChIP1 is mostly found in cholinergic interneurons (***Rhodes et al., 2004***; ***Xiong et al., 2004***).

Another possible way in which Kv4.2 channel function might be enhanced in HD iSPNs is by dephosphorylation. The Kv4.2 subunit is known to be phosphorylated by several TrkBR-activated serine-threonine protein kinases, like mitogen activated protein kinase (MAPK) (***Schrader et al., 2009***). Phosphorylation diminishes channel opening in response to depolarization. In HD iSPNs, TrkBR and MAPK signaling are impaired (***Plotkin et al., 2014***). Indeed, there was a decrease in phosphorylation of Kv4.2 subunits at Thr607 and Thr602 in HD striata (***Figure 4b***).

To test the possibility that TrkBR signaling regulated the assembly of Kv4 channel complexes, striatal slices were incubated with the TrkBR agonist – BDNF – and the ability of a Kv4.2 antibody to pull down KChIP subunits re-examined. In wild-type striata, BDNF significantly decreased the association of Kv4.2 subunits with KChIP2 (***Figure 4c***). However, in HD striata, BDNF did not significantly alter the association of Kv4.2 with KChIP2 (***Figure 4d***). Somewhat surprisingly, there wasn't a detectable difference in the association of KChIP2 and Kv4.2 protein in unstimulated, wild-type and HD striata (***Figure 4—figure supplement 1c***).

In iSPNs from symptomatic HD mice, TrkBR signaling is impaired by aberrant p75NTR signaling through rho-associated protein kinase (ROCK) (***Plotkin et al., 2014***). The selective ROCK inhibitor SR3850 (200 nM) (***Chen et al., 2008***) rescued the ability of BDNF to dissociate KChIP2 from Kv4.2 channels in HD striata (***Figure 4e***). These results are consistent with the proposition that TrkBR signaling normally leads to Kv4.2 subunit phosphorylation and the dissociation of KChIPs, which

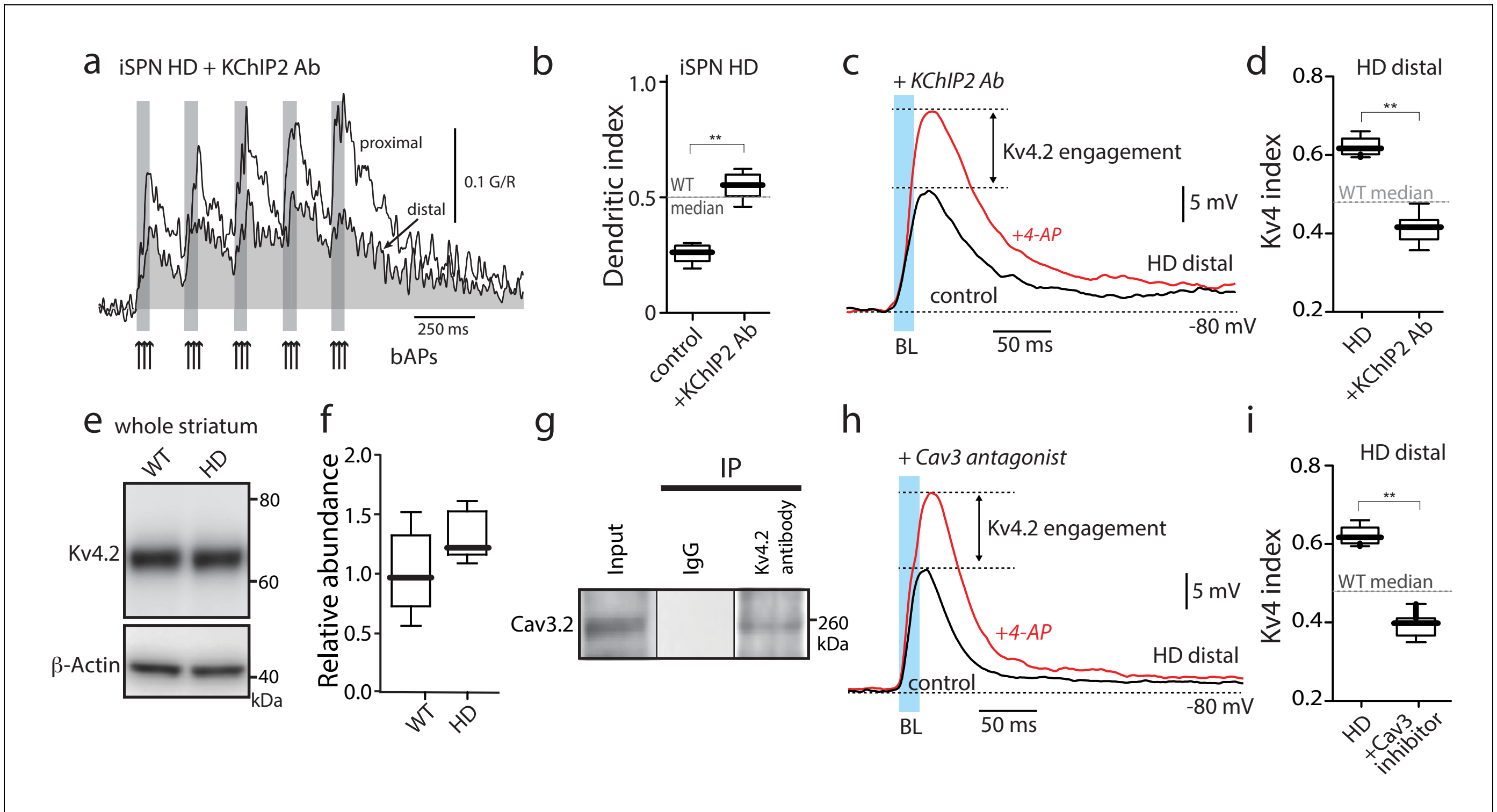


**Figure 3.** Functional upregulation of Kv4 channels depended upon KChIPs. (**a**) $Ca^{2+}$ transients in response to bAPs (arrows) at proximal and distal dendritic spines in iSPNs from BACHD mice with the internal perfusion of KChIP2 antibody (1:50); perfusion rescued dendritic excitability in HD iSPNs. (**b**) The dendritic index was significantly smaller with internal perfusion of the denatured KChIP2 antibody (boiled) (p=0.0022, Mann-Whitney U, Two-Tailed, n = 6). (**c**) Voltage responses to optogenetic stimulation of corticostriatal terminals at distal dendrites in iSPNs from BACHD mice with internal perfusion of KChIP2 antibody before and after 4-AP (2 mM). (**d**) The Kv4 index was significantly reduced by antibody perfusion (p=0.0006, Mann-Whitney U, Two-Tailed, n = 7). In all the dialysis experiments, recordings were taken >30 min after entering whole cell mode. (**e**) Western blot of Kv4.2 from striatum homogenates. (**f**) Kv4.2 protein levels show no significant difference between WT and Q175 ±mice (p=0.061, Mann-Whitney U, Two-Tailed, n = 5. (**g**) Co-immunoprecipitation of Cav3.2 channels with Kv4.2 channels from mouse striatum (Kv4.2 pulldown). (**h**) Voltage responses to optogenetic stimulation of corticostriatal terminals at distal dendrites of iSPNs from BACHD mice in the presence of the Cav3 $Ca^{2+}$ channel blocker mibefradil (1 µM) before and after 4-AP (2 mM). (**i**) Changes in the Kv4 index were consistent with the loss of dendritic excitability in iSPNs being mediated by the interaction of Cav3 $Ca^{2+}$ channels with Kv4.2 channels through KChIPs (p=0.0006, Mann-Whitney U, Two-Tailed, n = 7. See ***Figure 3—source data 1***.

DOI: https://doi.org/10.7554/eLife.40818.015

The following source data and figure supplements are available for figure 3:

**Source data 1.** Source data for ***Figure 3***.

DOI: https://doi.org/10.7554/eLife.40818.023

**Figure supplement 1.** Expression of Kv4 and KChIP mRNAs was unchanged in HD iSPNs.

DOI: https://doi.org/10.7554/eLife.40818.016

**Figure supplement 1—source data 1.** Source data for ***Figure 3—figure supplement 1***.

DOI: https://doi.org/10.7554/eLife.40818.017

**Figure supplement 2.** Colocalization of Kv 4.2 immunoreactivity and sparse-labeled ISPN dendrites.

DOI: https://doi.org/10.7554/eLife.40818.018

**Figure supplement 2—source data 1.** Source data for ***Figure 3—figure supplement 2***.

DOI: https://doi.org/10.7554/eLife.40818.019

**Figure supplement 3.** Pan-KChIP antibody normalized HD iSPN dendritic excitability.

DOI: https://doi.org/10.7554/eLife.40818.020

**Figure supplement 3—source data 1.** Source data for ***Figure 3—figure supplement 3***.

DOI: https://doi.org/10.7554/eLife.40818.021

**Figure supplement 4.** Cav1.3 channel was not associated with Kv4.2.

DOI: https://doi.org/10.7554/eLife.40818.022

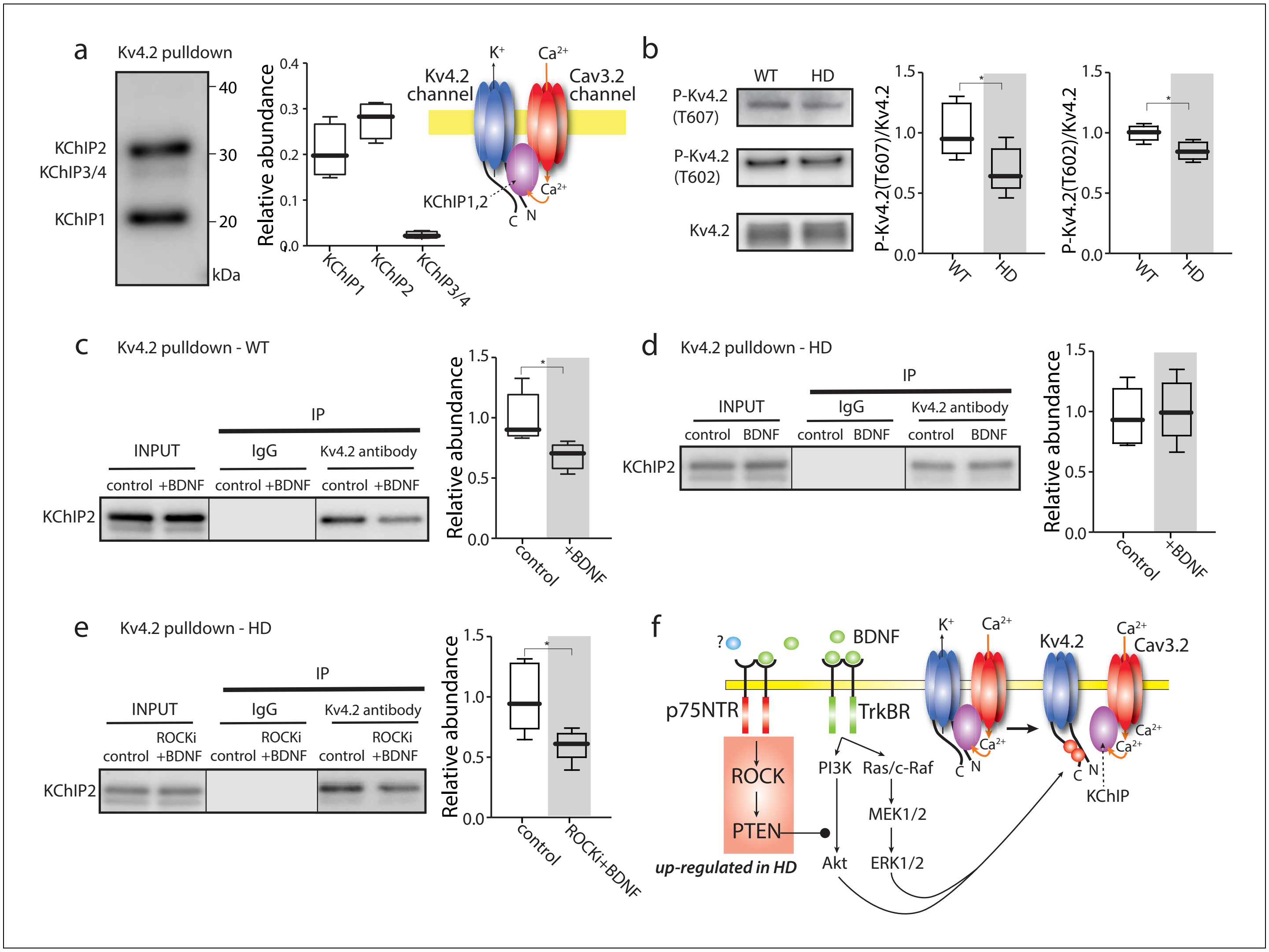


**Figure 4.** TrkBR signaling regulated Kv4 channel association with KChIPs. (**a**) KChIPs co-immunoprecipitated with Kv4.2 in mice striata, KChIP1 and KChIP2 more robustly associated with Kv4.2 than KChIP3/4. To the right is a schematic of the Kv4.2/KChIP/Cav3.2 membrane complex. (**b**) Detection of Kv4.2 phosphorylation at Thr607 and Thr602, P-Kv4.2 levels were normalized to total Kv4.2; in Q175 ±mice, Kv4.2 phosphorylation was decreased at both Thr607 (p=0.03078, Mann-Whitney U, Two-Tailed, n = 6) and Thr602 (p=0.0226, Mann-Whitney U, Two-Tailed, n = 6). (**c**) Co-immunoprecipitation of KChIP2 with Kv4.2 in WT striata after incubation with BDNF or with vehicle control; association of KChIP2 was decreased (p=0.01208, Mann-Whitney U, Two-Tailed, n = 5). (**d**) Co-immunoprecipitation of KChIP2 with Kv4.2 in Q175 ±striata after incubation with BDNF or with vehicle control; no difference in Kv4.2 association with KChIP2 (p=0.83366, Mann-Whitney U, Two-Tailed, n = 5). (**e**) Co-immunoprecipitation of KChIP2 with Kv4.2 in Q175 ±striata after incubation with ROCKi +BDNF or with vehicle control; association of KChIP2 with Kv4.2 was decreased (p=0.03662, Mann-Whitney U, Two-Tailed, n = 5). (**f**) Diagram of the proposed mechanism showing how TrkBR and p75NTR signaling modulate KChIP association with Kv4.2 and Kv4.2 channel gating. See ***Figure 4—source data 1***.

DOI: https://doi.org/10.7554/eLife.40818.024

The following source data and figure supplements are available for figure 4:

**Source data 1.** Source data for ***Figure 4***.

DOI: https://doi.org/10.7554/eLife.40818.027

**Figure supplement 1.** The co-immunoprecipitation experiments detected specific proteins.

DOI: https://doi.org/10.7554/eLife.40818.025

**Figure supplement 1—source data 1.** Source data for ***Figure 4—figure supplement 1***.

DOI: https://doi.org/10.7554/eLife.40818.026

decreases Kv4 channel opening (**Figure 4f**). In HD iSPNs, the impairment of TrkBR signaling appears to blunt this modulation increasing Kv4 channel opening and reducing dendritic excitability.

## Rescuing TrkBR signaling restored dendritic excitability

To test whether defective TrkBR regulation of Kv4 channels could account for the alterations in dendritic excitability in HD iSPNs, bAP-evoked dendritic $Ca^{2+}$ transients were measured in the presence and absence of the TrkBR agonist BDNF. In wild-type iSPNs, BDNF enhanced both the proximal and distal bAP evoked $Ca^{2+}$ transients (**Figure 5a,b**) and significantly increased the dendritic index (**Figure 5c**). In HD iSPNs, the ratio of distal to proximal dendritic excitability remained low in the presence of BDNF (**Figure 5d**). To complement the ROCK inhibitor experiments, two other strategies were used to restore TrkBR signaling in HD iSPNs (**Plotkin et al., 2014**). First, phosphatase and tensin homolog (PTEN) was knocked down using an AAV delivered shRNA (**Stavarache et al., 2015**); this effectively restored dendritic excitability to HD iSPNs (**Figure 5d–f**). Second, a TAT peptide pep5 (1 μM) targeting p75NTR was bath applied (**Ilag et al., 1999**; **Yamashita and Tohyama, 2003**); again, this intervention restored the dendritic excitability of HD iSPNs (**Figure 5f**).

To more directly probe TrkBR signaling in distal dendrites, visualized corticostriatal synapses were optogenetically stimulated with a spot laser and somatic EPSPs monitored (as described above). Based upon the co-immunoprecipitation experiments, inhibition of ROCK should enable TrkBR signaling, promote dissociation of KChIPs from Kv4 channels and decrease their activation. As expected, in wild-type iSPNs, BDNF (50 nM) increased the amplitude of cortically evoked EPSPs (**Figure 5g**). Bath application of the selective ROCK inhibitor (SR3850, 200 nM) alone had no effect on the Kv4 index (**Figure 5h**), in agreement with previous work suggesting little or no activation of this pathway in wild-type iSPNs (**Plotkin et al., 2014**). In contrast, in HD iSPNs, the ROCK inhibitor enabled BDNF to boost EPSP amplitude and to decrease the relative engagement of Kv4 channels, as estimated by Kv4 index (**Figure 5h**).

## Suppression of mHtt rescued iSPN excitability and synaptic plasticity

An unresolved question is whether the effects of mHtt on iSPNs are cell or regionally autonomous. Many neurons that innervate iSPNs, like cortical pyramidal neurons, manifest pathophysiology in models of HD (**Cummings et al., 2009**; **Virlogeux et al., 2018**). As a consequence, it could be that the adaptations in HD iSPNs are secondary to neuronal dysfunction elsewhere. To determine whether the dendritic deficit in HD iSPNs was dependent upon mHtt in the striatum, a zinc finger protein (ZFP) targeting the expanded CAG repeat in the mHtt gene was used. In heterologous expression systems, the mHtt ZFP reduced mHtt mRNA abundance, leaving wild-type mRNA unaltered (**Figure 6—figure supplement 1**). The ZFP expression construct and a non-binding ZFP (nbZFP, ΔBD), both with tdTomato reporter cassettes, were packaged in AAV (serotype 9/2) and injected into the striatum of 6 month old HD (Q175+/-) iSPN reporter mice (**Figure 6a**). Four months later, mice were sacrificed and striatal expression of mHtt and Htt mRNA assessed using qPCR. Roughly half (47%) of the DAPI-positive cells around the injection site expressed the tdTomato reporter (indicating AAV infection). In samples from this region, mHtt mRNA abundance was reduced whereas that for Htt was unaltered (**Figure 6b**). The nbZFP construct did not alter either mHtt or Htt mRNA abundance (**Figure 6b**). Also, cortical mHtt mRNA expression was unaffected by striatal delivery of the ZFP AAV (**Figure 6b**).

To assess the functional impact of diminishing mHtt expression, the dendritic excitability of iSPNs was assessed using a simplified bAP burst protocol (3 APs at 50 Hz). This simplified protocol was used because the first burst in the multi-burst protocol used in earlier experiments was always a reliable indicator of dendritic excitability (see **Figures 1–2**) and the shorter protocol maximized the quality and duration of recordings from iSPNs in slices from 10 month old mice. In wild-type iSPNs, infection with the ZFP construct had no discernible effect on dendritic morphology or dendritic excitability (**Figure 6c**). In HD iSPNs the nbZFP had no effect on distal dendritic excitability (**Figure 6c**). However, in HD iSPNs infected with the ZFP AAV, distal dendritic excitability was enhanced making their dendritic index indistinguishable from wild-type iSPNs (**Figure 6c,d**). These results suggest that the dendritic phenotype of HD iSPNs was a product of regional or cell autonomous factors.

Another consequence of impaired TrkBR signaling in HD iSPNs is a deficit in corticostriatal LTP (**Plotkin et al., 2014**). To determine if lowering mHtt with a ZFP could restore TrkBR-dependent LTP

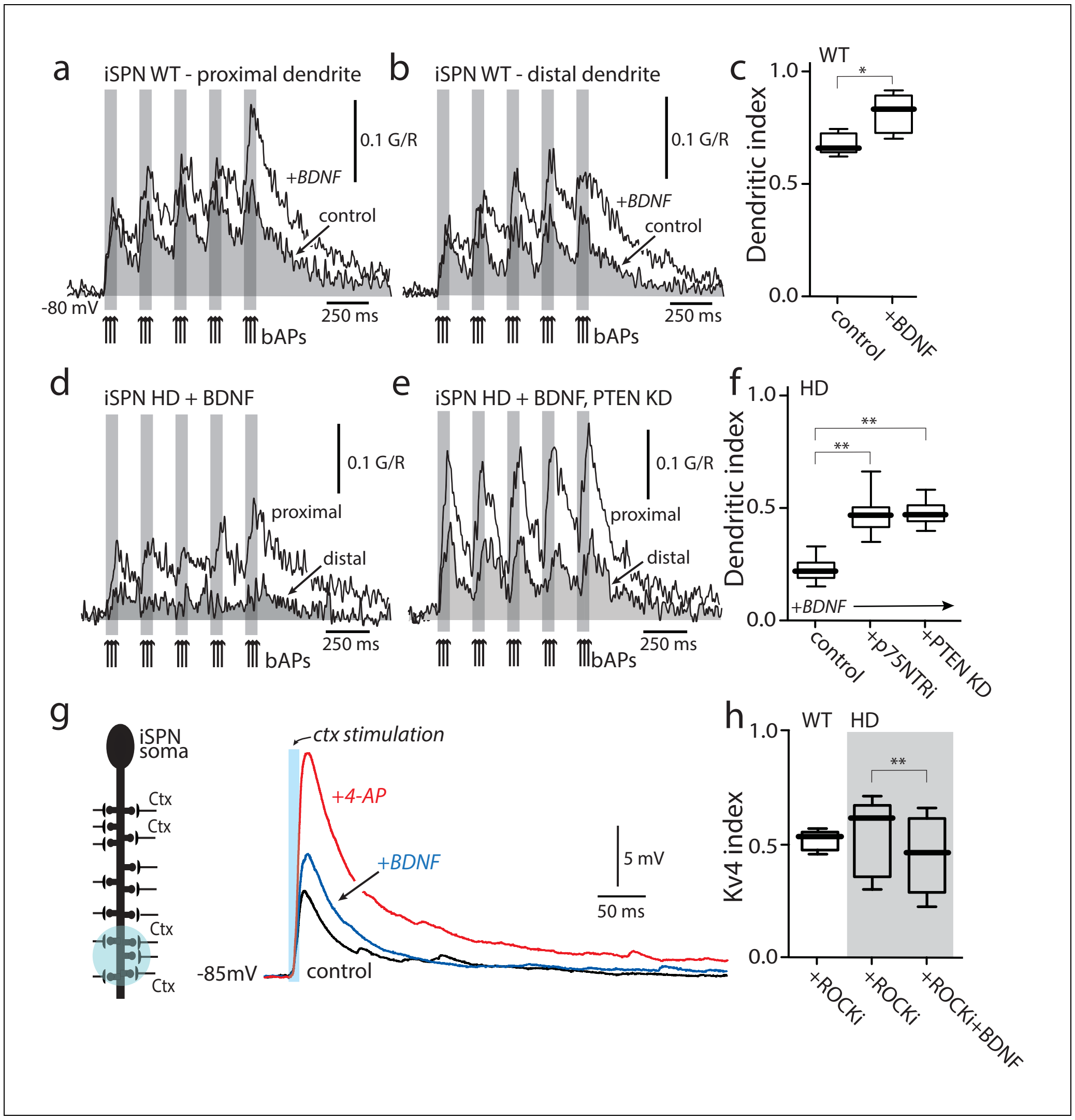


**Figure 5.** Enabling TrkBR signaling restored dendritic excitability. (**a**) $Ca^{2+}$ transients evoked by bAPs (arrows) at proximal and (**b**) distal dendritic spines in iSPNs from WT mouse in the presence of BDNF (40 ng/ml). (**c**) Dendritic index was significantly increased after BDNF application (p=0.00286, Mann-Whitney U, One-Tailed, n = 4). (**d**) Calcium transients in response to bAPs (arrows) in iSPNs from BACHD mouse in the presence of BDNF (40 ng/ml). (**e**) $Ca^{2+}$transients in response to bAPs (arrows) at proximal and distal dendritic spines in iSPNs from BACHD mouse injected with shRNA PTEN. Note that BDNF is able to rescue the loss of dendritic excitability. (**f**) Dendritic index demonstrates that the loss of dendritic excitability observed in iSPNs taken from BACHD mice can be attributed to the abnormal activity of p75 (BDNF (20 ng/ml) in the presence of p75 inhibitor pep5 (1 μM) and PTEN (traces not shown) (p=0.0012, Kruskal-Wallis three groups, n = 7). Note that in these experimental conditions BDNF is able to increase $Ca^{2+}$ transients in distal dendrites of iSPNs. (**g**) EPSPs evoked by optogenetic stimulation of distal dendrites of Q175 ±mice and their WT litter mates during bath application of the ROCK2 inhibitor (200 nM, black control trace), following 5 min BDNF (blue trace), and then in 2 mM 4-AP (red trace). (**h**) The boxplot shows a reduction in the Kv4 index in HD iSPNs following 5 min BDNF (50 nM) as estimated by Kv4 index (p=0.0245, Mann-Whitney U, One-Tailed, n = 9). See ***Figure 5—source data 1***.

DOI: https://doi.org/10.7554/eLife.40818.028

The following source data is available for figure 5:

**Source data 1.** Source data for ***Figure 5***.

DOI: https://doi.org/10.7554/eLife.40818.029

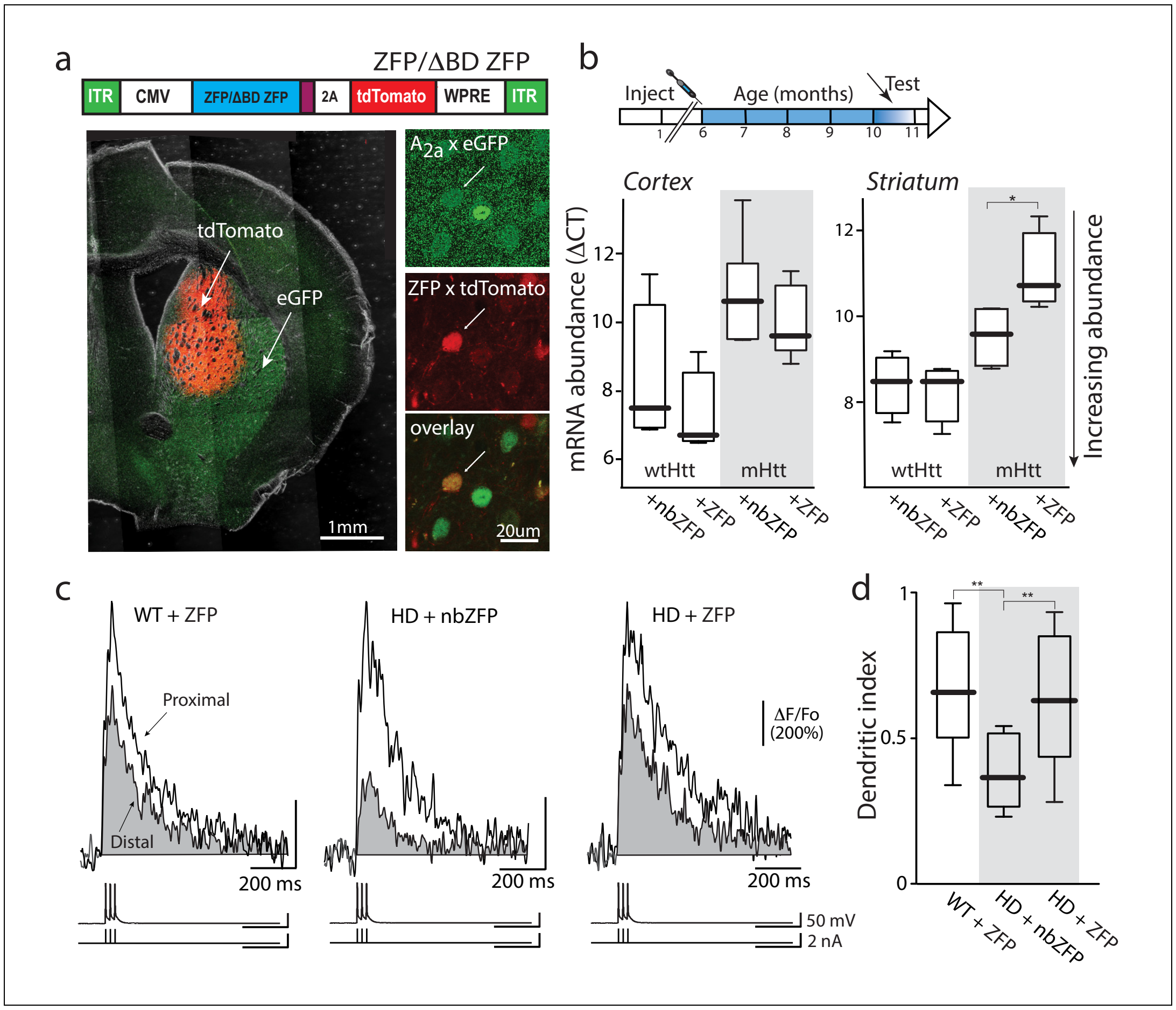


**Figure 6.** Suppression of mHtt normalized iSPN excitability. (**a**) ZFP vector map 1646 bp (EcoR1/HindIII) inserts from pVAX-30645 (Sigma) sub-cloned into pAAV-CMV-SV40-WPRE vector and packaged by Virovek. The expression of N-term NLS and C-term FLAG tagged human mutant Htt-repressor ZFP and tdTomato is bridged by viral 2A cleavage peptide. The vector shown here lacking the binding domain sequence (ΔBD, non-binding) was used as a vector control. The coronal slice images demonstrate both the coverage and restriction to the striatum of the stereotaxically injected AAV carrying ZFP and tdTomato genes (stereotaxic injection coordinates: ML = –1.7, AP = –0.98, DV = –3.6). High magnification images show that both iSPNs ($A_2A$ eGFP X Q175) and ZFP tdTomato infected cells can be clearly identified (center panel). These Q175 ±mice were injected at 6 months and tested at 10–11 months (cartoon). (**b**) mRNA abundance (ΔCT) levels were determined by qPCR in striata and cerebral cortex from Q175 ±mice 4 months after striatal delivery of AAV-ZFP. Striatal ZFP expression repressed mHtt gene expression in striatal but not cortical tissue (n = 4–5 mice; striatum: p=0.028, Mann-Whitney U, Two-tailed; cortex: p=0.73, Mann-Whitney U, Two-tailed). (**c**) 2PLSM line scans of bAP-evoked $Ca^{2+}$ transients, taken at proximal and distal locations within the same dendrite; shaft $Ca^{2+}$ transients were used to calculate the dendritic index (as previously defined). Left: $Ca^{2+}$ transients evoked in an iSPN from a WT mouse injected with AAV ZFP at 6 months and tested at 10 months; Center: Q175 ±iSPN $Ca^{2+}$ transients following striatal injection of nbZFP AAV; Right: Q175 ±iSPN $Ca^{2+}$ transients following striatal injection of ZFP AAV. (**d**) Summary boxplots showing the lack of effect of the nbZFP AAV on Q175 ±iSPN dendrites and restoration of normal excitability in Q175 ±iSPNs following ZFP AAV injection (WT littermates with ZFP: n = 5 mice, 10 cells. Q175 ±with nbZFP: n = 5 mice, 7 cells; p=0.0055, Mann-Whitney U, Two-tailed. Q175 ±with ZFP: n = 5 mice, 12 cells; p=0.0202, Mann-Whitney U, Two-tailed). See ***Figure 6—source data 1***.

DOI: https://doi.org/10.7554/eLife.40818.030

The following source data and figure supplements are available for figure 6:

**Source data 1.** Source data for ***Figure 6***.

DOI: https://doi.org/10.7554/eLife.40818.033

**Figure supplement 1.** ZFP-30645 construct selectively suppressed *mHtt* transcription in fibroblasts.

DOI: https://doi.org/10.7554/eLife.40818.031

**Figure supplement 1—source data 1.** Source data for ***Figure 6—figure supplement 1***.

DOI: https://doi.org/10.7554/eLife.40818.032

in symptomatic mice, the same strategy was used as described above; that is, HD (Q175+/-) mice and littermate controls were injected with ZFP-AAV and nbZFP-AAV at 6 months of age and then sacrificed 4–6 months later for electrophysiological analysis. As previously described (**Plotkin et al., 2014**), after dialysis with a $Cs^+$-containing solution to block $K^+$ channels, EPSPs were evoked by uncaging glutamate on 8–16 neighboring spines on distal dendrites of iSPNs (**Figure 7a**). In the presence of BDNF (50 ng/ml), an A2a adenosine receptor agonist (CGS21680, 200 nM), NMDA (5 µM) and glycine (5 µM), the somatic membrane was depolarized to −20 mV (1 s) four times, 'resting' for 10 s between depolarizations (**Figure 7b**). Then, the same spines were interrogated by glutamate uncaging at regular intervals (5, 10, 15 min) to determine whether there had been any lasting change in synaptic strength. In wild-type iSPNs, this protocol led to reliable LTP of axospinous synapses (**Figure 7c,d**). However, in iSPNs from HD mice infected with the vector containing the nbZFP construct, LTP was not observed, in agreement with previous work (**Plotkin et al., 2014**). However, in HD iSPNs infected with virus containing the ZFP construct, LTP was restored and was indistinguishable from wild-type iSPNs (**Figure 7c–e**). Thus, diminishing the mHtt burden with a ZFP restored TrkBR LTP at axospinous, glutamatergic synapses. Although the restoration of LTP was not dependent upon Kv4 $K^+$ channels, it confirmed that lowering mHtt in Q175 iSPNs rescued TrkBR signaling. These two consequences of TrkBR signaling – enhancement of dendritic excitability by reducing Kv4 $K^+$ availability and promotion of LTP – should work in concert in situ.

# Discussion

In two mouse models of HD (BACHD and Q175), the excitability of iSPN distal dendrites fell in parallel with the emergence of motor deficits (~6 months old). Our results suggest that the dendritic hypoexcitability in HD iSPNs stemmed from enhanced opening of Kv4 $K^+$ channels. This up-regulation was traced to an increased association between pore-forming Kv4 channel subunits, auxiliary KChIP subunits and Cav3 $Ca^{2+}$ channels. This association was negatively modulated by TrkBR signaling, which is impaired in HD iSPNs. Restoration of TrkBR function by inhibiting p75NTR signaling reversed the dendritic excitability deficits in HD iSPNs. As importantly, from a translational perspective, striatal expression of ZFPs targeting the expanded CAG repeat in *mHtt* also reversed the dendritic excitability deficit in tissue from symptomatic mice, suggesting that the HD phenotype of iSPNs was reversible and regionally, if not cellularly, autonomous. Thus, our studies not only provide new insight into how dendritic Kv4 channel function in iSPNs is regulated by TrkBR signaling and how this goes awry in HD, they suggest that a regional gene therapy could be of therapeutic value to symptomatic HD patients.

## Kv4 channel up-regulation induced dendritic hypoexcitability

Kv4 $K^+$ channels regulate the dendritic propagation of bAPs, as well as the response to local, synaptic depolarization in many neurons, including SPNs (**Day et al., 2008**; **Hoffman et al., 1997**; **Johnston et al., 2000**). By modulating bAP amplitude, Kv4 $K^+$ channels control opening of voltage-dependent $Ca^{2+}$ channels and dendritic $Ca^{2+}$ influx making bAP-evoked alterations in dendritic $Ca^{2+}$ concentration a suitable surrogate measure of Kv4 channel function (**Carter and Sabatini, 2004**; **Day et al., 2008**; **Häusser et al., 2000**; **Johnston et al., 2003**). To allow for comparisons of dendritic bAP-evoked $Ca^{2+}$ signals between neurons, the distal dendritic $Ca^{2+}$ signal was normalized by that taken in a parfocal proximal dendrite. In addition, the fluorescence of the $Ca^{2+}$ indicator was corrected for other optical variables by using the fluorescence of a red dye as a normalization factor ($\Delta G/R$). In iSPNs from both BACHD and Q175 HD mice, the ratio of distal to proximal bAP-evoked $Ca^{2+}$ transients were smaller than in age-matched, wild-type iSPNs; this deficit was corrected by knocking down the expression of Kv4.2 subunits or by pharmacologically inhibiting Kv4 channels with 4-AP. The up-regulation in distal dendritic Kv4 channels in HD iSPNs also was manifested in the response to locally evoked synaptic responses, as 2 mM 4-AP enhanced the amplitude of optically evoked EPSPs in the distal dendrites of HD iSPNs more than in wild-type iSPNs.

The up-regulation in Kv4 $K^+$ channel currents in HD iSPNs was not attributable to a change in the expression of Kv4 subunit genes in HD striata, but rather to their association with KChIPs. Previous work in other systems has shown that auxiliary KChIP subunits not only can enhance Kv4 channel trafficking to the membrane, but they can also increase the opening of Kv4 channels in response to depolarization (**An et al., 2000**; **Kunjilwar et al., 2004**; **Shibata et al., 2003**). Although the density

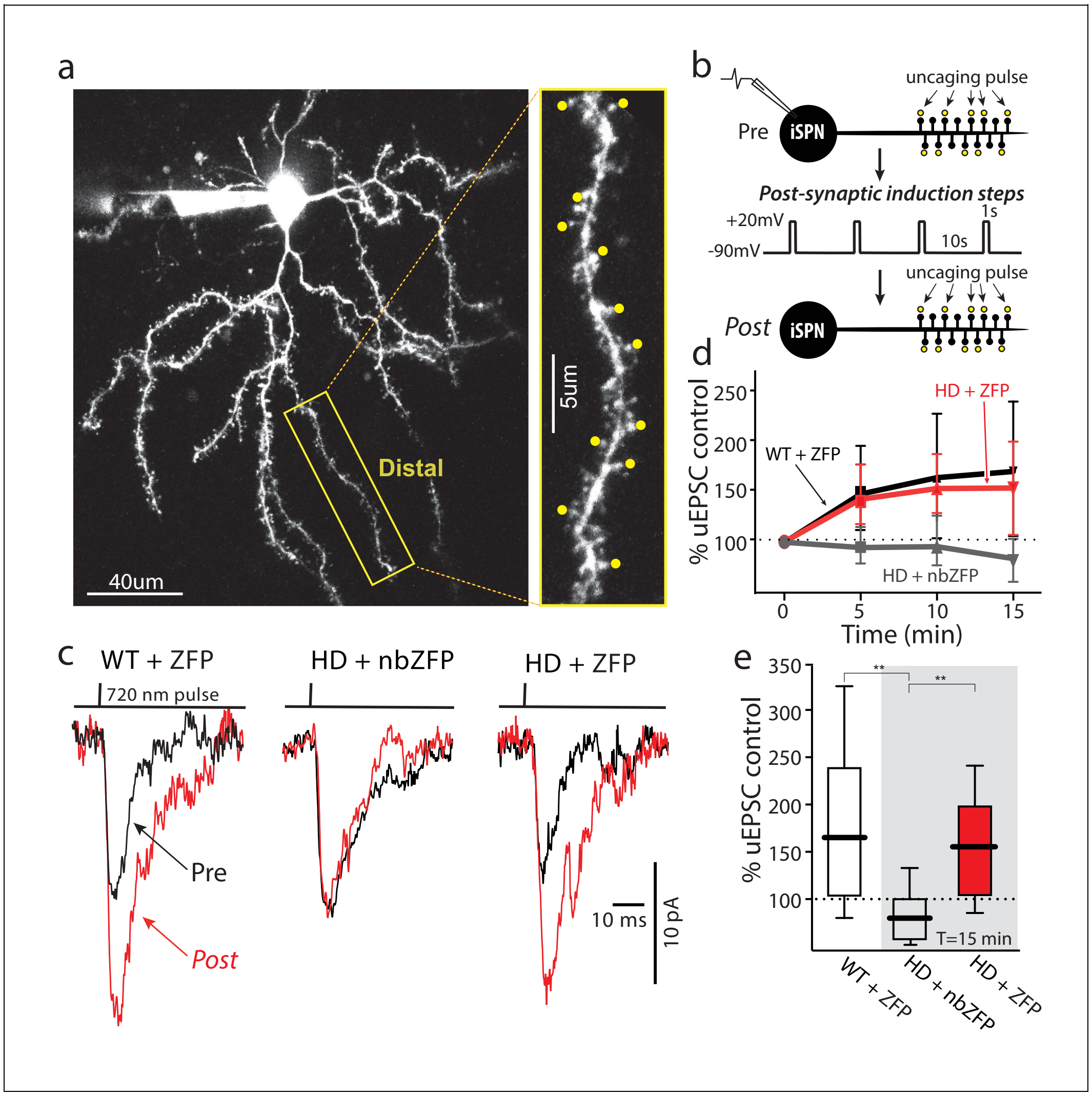


**Figure 7.** Suppression of mHtt restored TrkB receptor-mediated spine LTP in distal iSPN dendrites. (**a**) Maximum projection image of ZFP +Q175±iSPN with a high magnification image of a distal dendrite where 2PLSM spot uncaging of MNI-Glu was conducted (uEPSCs, yellow dots). (**b**) Schematic of the LTP induction protocol. (**c**) Traces (uEPSCs) showing individual uEPSCs before (*pre*, black) and after (*post*, red) the post-synaptic induction steps were evoked. (**d**) Time course showing the evolution of the TrkBR-mediated LTP normalized to pre-induction EPSC amplitude for ZFP +Q175+/- (red line), nbZFP +Q175+/- (grey line), and their ZFP +WT littermates (black line). (**e**) Summary boxplots constructed from the 15 min post-induction time point shows first, the loss of spine LTP in Q175 ±that received the nbZFP compared to their WT littermates that received the functional ZFP vector; and then second, the recovery of the spine LTP in Q175 ±that received the functional ZFP: ZFP +WT littermate (n = 4 mice, 4 cells, 32 spines), nbZFP +Q175+/- (n = 5 mice, 5 cells, 40 spines), and ZFP +Q175+/- (n = 5 mice, 6 cells, 48 spines); p=0.0001, Mann-Whitney U, Two-tailed. See ***Figure 7—source data 1***.

DOI: https://doi.org/10.7554/eLife.40818.034

The following source data is available for figure 7:

**Source data 1.** Source data for ***Figure 7***.

DOI: https://doi.org/10.7554/eLife.40818.035

of Kv4 immunoreactivity in the striatal neuropil made these experiments difficult, there was no discernible difference in the density of these channels in the proximal and distal dendrites of wildtype and Q175 iSPN dendrites – arguing against a primary role for trafficking. Nonetheless our data are consistent with the proposition KChIPs enhanced the function of Kv4 channels in HD iSPNs. Four observations support this conclusion. First, in striatal membrane extracts, Kv4.2 channels formed

protein complex with KChIPs and Cav3 channels (**Kunjilwar et al., 2004**; **Shibata et al., 2003**). Second, dialysis with a pan-KChIP or KChIP2 antibody that disrupts the interaction between Kv4.2 and KChIPs (**Anderson et al., 2010**) normalized the dendritic response to bAPs and to local synaptic stimulation. Finally, blocking $Ca^{2+}$ entry through Cav3 channels, which also complex with KChIPs and serve as a source of $Ca^{2+}$ to regulate the positive gating interaction of KChIPs on Kv4 channels (**Turner and Zamponi, 2014**), blunted the ability of Kv4 channels to modulate EPSPs evoked in distal dendrites.

Our studies found no evidence supporting the proposition that an alteration in Cav3 $Ca^{2+}$ channel expression or function contributed to the HD iSPN phenotype. The expression of $Ca^{2+}$ channels was not changed in Q175 iSPNs and Kv4 $K^{+}$ channel inhibition normalized the EPSPs produced by distal uncaging of glutamate. But, a causal role for Cav $Ca^{2+}$ channels cannot be completely excluded. Moreover, alterations in $Ca^{2+}$ signaling might have contributed to our observations in other ways. In HD models, there is increased ryanodine receptor (RYR) mediated release of $Ca^{2+}$ from intracellular stores (**Chen et al., 2011**; **Suzuki et al., 2012**). As RYR-mediated release of $Ca^{2+}$ from intracellular stores adds to the bAP-evoked dendritic $Ca^{2+}$ signal in iSPNs (**Plotkin et al., 2013**), it is possible that boosting this signal augmented KChIP-dependent facilitation of Kv4 channel opening.

## Kv4 and KChIP association was regulated by TrkBR signaling

Although it was evident that an increased association of KChIP proteins with Kv4 channel subunits was responsible for the suppression of dendritic excitability in HD iSPNs it was less clear what caused the change. There was no detectable increase in the expression of any KChIP mRNA in HD striata. One other well-known regulator of Kv4 channel gating is phosphorylation (**Hoffman and Johnston, 1998**; **Yuan et al., 2002**). The C-terminal region of the Kv4.2 subunit has several sites that are phosphorylated by serine/threonine kinases like ERK (**Schrader et al., 2009**; **Schrader et al., 2006**). Phosphorylation of these sites decreases Kv4 channel opening in response to depolarization by increasing steady-state inactivation. Association of Kv4 subunits with KChIPs is necessary for the effects of phosphorylation on gating to be seen (**Schrader et al., 2006**). Given that phosphorylation and KChIP association have the opposite effect on Kv4 gating, it was possible that the effects of phosphorylation were mediated by dissociation of the KChIP subunit. While KChIPs bind to the N-terminal region of Kv4 subunits (**Gulbis et al., 2000**; **Sewing et al., 1996**), this region is projected to come in close physical proximity to the C-terminal phosphorylation sites in a native channel (**Jerng et al., 2004**; **Kim et al., 2004**) lending some plausibility to the hypothesis. Indeed, in wild-type striata, stimulation of TrkBRs with BDNF significantly decreased the association of Kv4 channels with KChIPs and increased dendritic excitability in iSPNs measured either with bAPs or local synaptic stimulation.

This inference provides a ready explanation for why Kv4 channels in HD iSPN dendrites were more readily engaged by depolarization. Previous work has shown that TrkBR signaling is blunted in HD iSPNs (**Brito et al., 2014**; **Plotkin et al., 2014**). Consistent with these previous results, phosphorylation of Kv4.2 subunits at two C-terminal sites (Thr607, 602) was reduced in HD striata. Moreover, BDNF stimulation of TrkBRs in HD striata failed to alter the association of KChIPs and Kv4.2 subunits, unlike wild-type striata. The deficit in TrkBR signaling in iSPNs is attributable to up-regulation in p75NTR signaling through ROCK and PTEN, not to any deficit in TrkBR expression or coupling to Ras/c-Raf or PI3K (**Brito et al., 2014**; **Plotkin et al., 2014**). Inhibition of ROCK rescued the ability of BDNF and TrkBR signaling to dissociate Kv4.2 subunits from KChIPs in HD striata. Moreover, ROCK inhibition restored the ability of TrkBR signaling to increase the response to synaptic stimulation in HD iSPNs. Lastly, inhibiting other elements in the p75NTR signaling cascade – p75NTRs themselves or PTEN – also rescued dendritic excitability in HD iSPNs following exposure to BDNF. All these lines of evidence support the conclusion that the down-regulation of dendritic excitability in HD iSPNs was mediated – at least in part – by the failure of TrkBR signaling to promote phosphorylation of Kv4.2 subunits and dis-association of KChIPs. Thus, the disruption in TrkBR signaling in HD models results not only in impaired potentiation of excitatory synapses, but in dendritic hypoexcitability. This coordinated deficit should make it more difficult to appropriately activate the basal ganglia indirect pathway and suppress unwanted movements providing a mechanistic foundation for early hyperkinetic features of the disease (**Albin et al., 1992**). mHtt effects were reversible and regionally autonomous

Why iSPNs are preferentially vulnerable in human HD patients remains unresolved. It has been hypothesized that iSPN vulnerability in HD is actually secondary to the failure of cortical neurons to synthesize and transport BDNF to the striatum, which is necessary for normal iSPN function (**Virlogeux et al., 2018**; **Zuccato et al., 2010**). While a transport deficit may contribute to the phenotype in mouse HD models there are striatal TrkBR signaling deficits that precede any discernible reduction in cortically-derived BDNF (**Brito et al., 2013**; **Plotkin et al., 2014**). It also is difficult to understand how a global deficit in cortical delivery of BDNF to the striatum might selectively alter iSPN excitability; neighboring dSPNs also express TrkBRs and there is no evidence that these two SPN populations are innervated by different cortical neurons (**Kress et al., 2013**; **Lobo et al., 2010**). However, these two types of SPN have a wide range of intrinsic differences that could underlie their disparity in sensitivity to mHtt expression (**Gerfen and Surmeier, 2011**; **Heiman et al., 2014**).

In support of the proposition that there is regional autonomy to the HD phenotype striatally-delivered ZFPs targeting the CAG repeat domain of mHtt reversed the dendritic excitability deficits in HD iSPNs. These experiments also demonstrate that the consequences of mHtt expression on striatal physiology are reversible, at least in the near-term, in agreement with previous work arguing that the continuous supply of mutant protein was necessary to maintain the disease phenotype (**Yamamoto et al., 2000**).

A key translational question is whether therapeutic strategies targeting a single region of the basal ganglia or cerebral cortex will be able to ameliorate the HD symptoms, in spite of the widespread expression of mHtt. The circuits most affected in HD are synaptically coupled (**Plotkin and Surmeier, 2015**) making it possible that a deficit in one node of the network is responsible for pathophysiology elsewhere. If this were the case, regionally-targeted gene therapies would become an option for patients. The demonstration that regionally autonomous mechanisms control the iSPN HD phenotype appear to make the striatum a necessary therapeutic node. However, it is not clear that this node will turn out to be sufficient (**Wang et al., 2014**).

# Materials and methods

## Brain slice preparation

Transverse corticostriatal or parasagittal slices (300 or 275 µm thickness) were obtained as previously described (**Kawaguchi et al., 1989**) from PD80-90 for asymptomatic animals and PD180-210 for symptomatic mice. The following transgenic male mice were used: $D_2$BAC-EGFP (FVB, RRID: MMRRC_000230-UNC), $D_1$BAC-EGFP (FVB, RRID:MMRRC_000297-MU), BACHD (FVB, RRID:IMSR_JAX:027433) x $D_2$BAC-EGFP (FVB), BACHD (FVB) x $D_1$BAC-EGFP (FVB), Thy1-ChR2 (C57BL/6, RRID: IMSR_JAX:007612), Thy1-ChR2 (C57BL/6) X BACHD (FVB), $A_2$A eGFP (C57BL/6, RRID:MMRRC_010541-UCD, backcrossed to C57Bl/6J in the Surmeier lab) x Q175 (C57BL/6, RRID:IMSR_JAX:029928), Thy1-ChR2 (C57BL/6) x $A_2$A eGFP (C57BL/6) x Q175 (C57BL/6). All procedures conformed to the guidelines of Northwestern University Animal Care and Use Committee. Mice were acutely anesthetized with a mixture of ketamine (50 mg/kg) and xylazine (4.5 mg/kg) and perfused transcardially with oxygenated ice-cold saline (4˚C) containing in mM: 125 NaCl, 2.5 KCl, 1 $MgCl_2$, 2 $CaCl_2$, 25 $NaHCO_3$, 1.25 $NaH_2PO_4$ and 25 glucose (saturated with 95% $O_2$-5% $CO_2$; pH 7.4; 298 mosM/l). After perfusion, mice were decapitated and brains rapidly removed. Slices were obtained in oxygenated ice-cold saline using a vibratome (VT1000S, Leica Microsystems). Slices were then transferred to saline at room temperature (21–25˚C) where they remained for 1 hr before recordings.

## Electrophysiological recordings

For electrophysiological recordings slices were transferred to a submersion-style recording chamber mounted on an Olympus BX51 upright microscope (60X/0.9 NA objective) equipped with infrared differential interference contrast. Whole-cell patch clamp electrophysiological recordings were performed with Multiclamp 700B amplifier. Signals were filtered at 1 KHz and converted to digital format with Digidata 1400. Stimulation and display of electrophysiological recordings were obtained with freeware WinFluor (John Dempster, Strathclyde University, Glasgow, UK. http://spider.science.strath.ac.uk/sipbs/software_winfluor.htm) that synchronizes two-photon imaging and electrophysiology. Targeted electrophysiological recordings were obtained from iSPN or dSPN. Patch pipettes (4–6 MΩ) were prepared with a Sutter Instruments horizontal puller using borosilicate glass with

filament and filled with (in mM): 135 $KMeSO_4$, 5 KCl, 5 HEPES, 0.05 EGTA, 2 ATP-$Mg_2$, 0.5 GTP-Na, 10 phosphocreatine-di (tris); pH was adjusted to 7.25 with KOH and osmolarity to 270–280 mosM. To record $Ca^{2+}$ transients in dendritic spines and shafts, cells were filled with 100 µM Fluo-4 pentapotassium salt and 25 µM Alexa Fluor 568 hydrazide $Na^+$ salt (Invitrogen). Electrophysiological characterization of neurons was made in current clamp configuration. The amplifier bridge circuit was adjusted to compensate for electrode resistance. Access resistances were continuously monitored and experiments were discarded if changes > 20% were observed. Digitized data were imported for analysis with commercial software (IGOR Pro 6.0, WaveMetrics, Oregon).

## Two-photon $Ca^{2+}$ imaging

$Ca^{2+}$ transients were evoked by back propagating action potentials (bAPs) (five bAP triplets, each; 50 Hz intra train; 5 Hz between trains; in the ZFP experiments, bAPs were generated with a single 50 Hz triplet). $Ca^{2+}$ transients were recorded at distal dendritic spines (>100 µm) and proximal dendritic spines (50 ~ 60 µm), each from co-planar sections of the same dendrite using a laser scanning microscope system (Ultima, Bruker Technologies; formerly Prairie Technologies) with a tunable imaging laser (Chameleon-Ultra 1, Coherent Laser Group, Santa Clara, CA) set to 820 nm and Olympus 60X/1.0NA water-dipping objective lens. Line scan signals were acquired with ~0.2 µm pixel and 10 µs/pixel dwell time. Red signals from Alexa Fluor 568 (580–630 nm; R3896 PMT, Hamamatsu) were used to visualize dendrites, whereas green signals from Fluo-4 (490–560 nm) were used to record $Ca^{2+}$ transients. Background subtraction of the Hamamatsu H7422P-40 GaAsP (green channel) PMT signal is critical since the Fluo-4 fluorescence is low with basal [$Ca^{2+}$] *and* these PMTs have higher background voltage which varies with the applied high voltage (700V to 900V). The mean fluorescence reported as a function of time ($F(t)$) was the spatial average of 5–15 adjacent line scan pixels, while the basal fluorescence, $F_o$, was the average of the first 30-time points within a line scan. Due to indicator buffering, the total net change in calcium current is represented by the area of the fluorescence response above basal levels: $(F(t) – F_0) / F_0$. The red channel fluorescence was used to normalize the $Ca^{2+}$ signals obtained from proximal and distal sections. For the ZFP experiments using the triplet bAP, IGOR Pro (WaveMetrics, Lake Oswego, OR) was used for data smoothing and statistics.

## Two-photon laser uncaging

Simultaneous two-photon $Ca^{2+}$ imaging (820 nm) and two-photon laser uncaging (720 nm) were performed using an additional laser coupled to a second, separate, independently controlled galvanometer mirror pair (Ultima, Bruker Corporation). The two laser beams are optically combined (760DCLPXR, Chroma Technologies) in the scan head on their way to the objective lens and sample plane. MNI-glutamate (5 mM) was superfused in the recorded area and excited by the photo-stimulation laser (Chameleon, Coherent Laser Group, Santa Clara, CA). Pulses of 1 ms duration (~10 mW sample power) were delivered to single spines located in the same focal plane (5–10 spines) where the laser power was calibrated to evoke a somatic excitatory postsynaptic potential of 1–2 mV for each spine. Custom written software (*WinFluor*, John Dempster and its *PhotoStimulusEditor* module, Nicholas Schwarz; features now available in *PrairieView* 5) was used to direct, control, test, synchronize, and record $Ca^{2+}$ transients, electrophysiological recordings, and laser stimulation.

## Single spine synaptic potentiation

iSPNs were held at −90 mV in the presence of an ''LTP cocktail'' containing 50 ng/ml BDNF, 5 µM NMDA, 5 µM glycine, 1 µM TTX, 200 nM CGS21680, 1 µM sulpiride, 50 µM CPCCOEt, 1 µM MPEP, and 0.1% BSA, dissolved in HEPES-buffered ACSF. This LTP cocktail was superfused (0.4 ml/hr) over the slice using a syringe pump and multibarreled perfusion manifold (Cell MicroControls). Baseline uEPSCs were recorded from approximately eight coplanar distal (greater than 100–120 um from soma) spines, separated by 200 ms, and averaged two times (5 s inter-trial interval). The cell was then stepped from −90 to +20 mV for 1 s, four times total, 10 s between steps. uEPSCs were then measured from the same spines in the same manner 5, 10, and 15 min after the depolarizing voltage steps. Unless otherwise indicated, represented time points are 15 min postinduction. Series resistance was monitored, and recordings with a change of more than 20% were rejected from analysis.

## KChIP antibody dialysis

KChIPs antibodies were perfused through the patch pipette to the interaction between KChIPs and Kv4 subunits (***Anderson et al., 2010***). The broad-spectrum antibody (pan-KChIP, Neuromab 75–006) was diluted 1:50 in the pipette solution. The KChIP2 antibody (Neuromab 75–004) also was diluted at the same 1:50 ratio. As a control, the KChIP2 antibody was boiled and diluted (1:50). Antibodies were dialysed for 30 min before conducting physiological assays.

## Optogenetic visible laser stimulation

Simultaneous two-photon $Ca^{2+}$imaging (820 nm) and ChR2 optogenetic photo-stimulation (473 nm) were performed with an additional blue laser (Aurora laser launch, Prairie Technologies) sharing the same galvanometer pair and software control as the 720 nm laser. The Point Photo-activation (Prairie Technologies) with the 60x/1.0 objective can deliver sub-µm (small spot) or ~10 µm (large spot) diameter photo-stimulation. The targeted 473 nm spots were positioned adjacent to individual dendritic spines to photo-stimulate presynaptic terminals impinging on iSPNs; in this way, 5–10 individual spines lying in the same focal plane could be stimulated. The laser power was also calibrated to evoke a somatic excitatory postsynaptic potential of 1–2 mV for each stimulated spine (small spot) or ~10 mV for 5–10 simultaneous stimulated spines (large spot). For the ROCK2 inhibitor experiments the large spot was used.

## Stereotaxic viral gene delivery

Stereotaxic injections of adeno-associated viral vector (AAV) carrying Kv4.2 shRNA, or shRNA PTEN +mCherry or control sequences were made in striatum (ML = −1.7, AP = 0.98, DV = −3.6) of isoflurane anesthetized 6-week-old C57BL/6 mice. Mice were allowed to recover for one-month post-injection.

## Quantitative real-time PCR analysis of Kv4.2 and Htt mRNA Expression

RNA was isolated using RNAeasy kit (Qiagen) from ST HDH Q7/111 cells and striatal tissue of mice injected with Kv4.2 shRNA or ZFP AAV. The RNA was reverse transcribed with Superscript III RT enzyme (Life Technologies) or qScript cDNA synthesis kit (Quanta Biosciences). Quantitative real-time PCR was performed using an ABI StepOnePlus Real Time PCR system with SYBR-Green PCR Master Mix (Applied Biosystems, Forster City, CA). The relative abundance of different transcripts was assessed by SYBR quantitative PCR in triplicate.

The following primers (Integrated DNA Technologies) were used for PCR amplification: GAPDH (accession number: NM_008084.2) mGAPDH-F ′-CATTTGCAGTGGCAAAGTGG-3′, mGAPDH-R 5′-GAATTTGCCGTGAGTGGAGT-3′; Kv4.2-F 5′- GTGTCGGGAAGCCATAGAGGC-3′, Kv4.2-R 5′- TTACAAGGCAGACACCCTGA-3; wild-type-mouse Htt_fw: CAG GTC CGG CAG AGG AAC C, Mut-mouse-Htt_Q175_fw: GCC CGG CTG TGG CTG A, Mut and wild-type Htt_rv*: TTC ACA CGG TCT TTC TTG GTG G (*wild-type and mutant Htt share the same reverse primer sequence (Htt gene accession number: NM_010414.3)). Experimental Ct values were normalized to GAPDH values using the formula: ΔCt = Ct (Kv4.2 or Htt) - Ct (GAPDH). The final expression levels were shown as ΔCt values. The PCR products were verified by melt curve analysis and agarose gels.

## AAV-Kv4.2 shRNA expression vector

To silence Kv4.2 expression, we have selected a target sequence (GCAAGAACTCAGTACAATT) (position 1849–1867) from the nucleotide sequence of mouse Kv4.2 (accession No. NM_019697.3) and a non-targeting control sequence (AGGATCAAATTGATAGTAAACC). The uniqueness of selected target sequences was confirmed by NCBI blast search. The pFB-cre-off-U6-CMV-EGFP-WPRE backbone vector was created by digesting pFB-CMV-AAV-SV40 plasmid (Virovek, Hayward, CA) with MluI and KpnI restriction enzymes to excise CMV promoter and ligated to 2.4 kb XbaI and SalI fragment containing floxed-U6-CMV-EGFP from pSicoR plasmid (Addgene plasmid 11579). To construct shRNA vectors, two pairs of oligonucleotides containing the antisense sequence, hairpin loop region (TTCAAGAGA), and sense sequence with cohesive BamHI and XhoI sites were synthesized (Integrated DNA Technologies Inc., Coralville, IA) and cloned into pAAV-floxed-U6-CMV-EGFP vector downstream to U6 promoter. The pFB-cre-off-U6-Kv4.2-CMV-EGFP-WPRE and pFB-creoff-

U6-control-shRNA-CMV-EGFP-WPRE constructs were packaged into rAAVs of serotype 9 (Virovek, Hayward, CA).

## Fluorescence-activated cell sorting and Cav3/Kv4 qPCR

Experimental procedures and data processing for fluorescence-activated cell sorting (FACS) and quantitative PCR (qPCR) were similar to the analysis in our previous studies (**Plotkin et al., 2014**). Ex vivo slices were prepared as described above, striata were microdissected, and single-cell suspensions were generated using a combination of enzymatic and mechanical dissociation procedures. Direct-pathway SPNs (dSPNs) and indirect-pathway SPNs (iSPNs) were purified using FACS on a cell sorter (BD Biosciences), based on tdTomato or enhanced green fluorescent protein (eGFP) expression. Dead cells were excluded based on the DAPI or propidium iodide labeling. Approximately 10,000 cells from each mouse were collected, lysed using RealTime ready Cell Lysis Buffer (Roche), stored at −80˚C, and used for qPCR analysis. Immediately following total RNA isolation using the RNAeasy Micro Kit (Qiagen), first-strand cDNA synthesis was performed using cDNA Supermix (Quanta Biosciences). cDNAs were then frozen at −20˚C until used. Desalted primers were custom synthesized (Invitrogen) and intron spanning whenever possible. qPCRs were run in triplicate using Fast SYBR Green Mastermix (Applied Biosystems) on a qPCR instrument (Roche Applied Science). Negative controls for contamination from extraneous and genomic DNA were run for every gene target. The cycling protocol consisted of 95˚C for 5 min followed by 45 cycles at 95˚C for 10 s, 60˚C for 10 s, and 72˚C for 10 s. The PCR cycle threshold (CT) values were determined by the maxima of the second derivative of the fluorescence response curves. Melting curves were performed to verify the amplification of single PCR products. A weighted CT was calculated from seven reference genes for each cell sample based on their expression stability for normalization. The mRNA levels in each group of samples were characterized by their median values. Quantification of transcript expression changes was performed using the ΔΔCT method. Results were presented as fold changes relative to the median of their respective wild type controls.

## Co-Immunoprecipitation and western blotting

wild-type or Q175 het mice of 6 to 8 month-old were isoflurane-anesthetized and sacrificed by cervical dislocation followed by decapitation. Coronal brain slices (350 μm) were prepared (Leica VT1200S) and allowed to recover in normal ACSF for 30–60 min (34˚C) with carbogen bubbling. In some experiments, brain slices were incubated in ACSF containing 50 ng/ml BDNF (Tocris) for 20 min, or 200 nM ROCK inhibitor SR3850 (CHDI Foundation) for 20 min followed by 50 ng/ml BDNF. Dorsal striatum tissues were then collected under dissection microscope and homogenized in Lysis buffer (Thermo Scientific) supplemented with Protease inhibitors (Halt Protease and Phosphatase Inhibitor Cocktail, Thermo Scientific). After centrifugation (16,000 g 10 min), supernatants containing 200 μg membrane protein were incubated with 5 μg goat anti-Kv4.2 antibody (C20, Santa Cruz Biotechnology), 5 μg rabbit anti-Kv1.2 antibody (ThermoFisher PA5-23006), or 5 μg anti-panKChIP antibody (Neuromab 75–006), or, as IP negative control, 5 ug of normal goat IgG (Thermo Scientific) overnight at 4˚C with constant mixing. Co-immunoprecipitation was carried out using Pierce Classic Magnetic Co-IP Kit (Thermo Scientific) following the instructions. Briefly, the mixtures were incubated with protein A/G magnetic beads at room temperature for 1 hr to bring down the antigen-antibody complexes. The beads were then washed two times with Pierce IP wash buffer and once with pure water. Immunocomplexes were eluted using 60 μl elution buffer and the low pH neutralized by adding neutralization buffer. Samples were prepared in NuPAGE LDS Sample Buffer (Invitrogen) and separated by SDS-PAGE. Primary antibodies to pan-KChIP (Neuromab 75–006, 1:1000), KChIP2 (Neuromab 75–004, 1:1000), Cav3.2 (Alomone ACC-025, 1:200), Cav1.3 (gift from Amy Lee, 1:1000), phospho-Kv4.2 (Thr607, Santa Cruz sc-377545, 1:200), phospho-Kv4.2 (Thr602, Santa Cruz sc-377574, 1:500), Kv1.2 (Neuromab 75–008, 1:500) and Kv4.2 (Neuromab 75–016, 1:1000) were used for Western blotting. Target protein bands were visualized with horseradish peroxidase-conjugated secondary antibodies (Cell Signaling) and enhanced chemiluminescent reagent SuperSignal West Dura (Thermo Scientific). Intensities of the target protein bands were detected by Odyssey Fc (Li-Cor) and quantified in Image Studio (Li-Cor).

## ZFP constructs

The plasmid (pVAX −30645) containing allele selective ZFP repressor cDNA and non-binding ZFP cDNA (CHDI-90001486) were obtained from Sigma and CHDI. 1646 bp ZFP and the 1180 bp non-binding ZFP inserts were subcloned and packaged into AAV9 by Virovek (Hayward, CA). The expression of N-term NLS and C-term FLAG tagged human mutant Htt-repressor ZFP and tdTomato is bridged by viral 2A cleavage peptide (***Figure 7***).

## Cell line

STHdh Q7/111 was obtained from Coriell Institute for Medical Research's HD Community Biorepository (Coriell ref# CHDI-90000072; RRID:CVCL_DN41). STHdh Q7/111 is a striatal derived cell line from a knock in transgenic mouse containing heterozygous Huntingtin (Htt) loci with a humanized Exon 1 with 111 polyglutamine repeats. The status of mycoplasma contamination was not reported. Cells were plated in 12 well plates and then infected with AAV carrying ZFP-30645 or nonbinding ZFP after overnight incubation and harvested after 72 hr of infection for RNA isolation.

## ZFP mediation suppression of mHtt in A2a-Q175 HETs

Stereotaxic injections of AAV carrying ZFP and tdTomato genes were made in striatum (ML = −1.7, AP = 0.98, DV = −3.6) of isoflurane-anesthetized 4-month-old A2a-EGFP and A2a-EGFP/Q175 mice. Mice were allowed to recover for at least 2 months post-injection. Q175 Mice and their WT littermates were randomly chosen from a large breeding colony comprised of multiple litters based on appropriate age. All mice were ear-tagged and the injections given and recorded by a technician. After recording and data analysis, the tag numbers were matched to the data sample so the data could be assigned to the appropriate test group.

## Image analysis of striatal coronal sections

Coronal sections spaced 100 μm apart comprising rostral, middle and caudal levels of the striatum were examined with confocal microscope (Fluoview Olympus). Confocal images of the entire slices were created under low power while 60X images were taken to visualize individual cells. For each coronal section, the entire three-dimensional stack of images from the ventral surface to the top of the section was obtained by using the Z drive and subsequently collapsed. Quantitative analyses of SPN infection rates were performed using the ImageJ (National Institutes of Health, Bethesda, MD, USA). EGFP (iSPNs), tdTomato (ZFP) and blue (DAPI) cells were counted from the confocal images to determine the percent of ZFP-tdTomato expressing cells and the percent of eGFP expressing iSPNs from the total DAPI stained cells.

## Quantitative analysis of 3d dendritic colocalization of kv 4.2 channels

Stereotaxic injections of AAV-EF1-Flex-tdTomato were made in striatum (ML = 1.9, AP = 0.98, DV = −3.8) of isoflurane anesthetized 6–8 months old A2a-Cre/Q175 mice and wild-type littermates. Three weeks later, mice were transcardially perfused with saline followed by freshly prepared 4% formaldehyde in 0.1 M Na-phosphate buffer, pH 7.4. Free-floating 80 μm thick sections were obtained using a vibratome and processed for immunolabeling. Sections were blocked in solution containing 5% goat serum and 0.3% Triton X-100 in PBS, pH 7.4 for 1 hr at room temperature. Sections were then labeled overnight at 4˚C with primary antibody to Kv4.2 (Neuromab 75–361, 1:100). Following three 10 min washes in PBS, sections were incubated with Alexa Fluor 568 secondary antibody (Invitrogen) at 1:400 for 30 min. Sections were rinsed and mounted with Vectashield (Vector Labs, R.I. 1.45). Confocal images were captured using Nikon A1R confocal laser microscope system with 100 x objective (N.A. 1.45). Z-stacks were deconvolved using AutoQuant 3 (Media Cybernetics) with Adaptive PSF (blind) method and theoretical PSF settings. Image were processed in Imaris Image Analysis (Bitplane). Threshold cutoff was used for background substraction in green channel (Kv 4.2 labeling). New surfaces were added to the volume rendered of proximal or distal dendrites labeled with tdTomato to create masked channel. ImarisColoc with automatic selection of the thresholds (for details see ***Costes et al., 2004***) was used to determine and quantify the colocalization. Random images generated by smoothing white noise with a Gaussian PSF of the specified width that have 0 or one correlation where excluded from the analysis.

## Pharmacological reagents

Stock solutions were prepared before experiments and added to the perfusion solution or focally applied with pressure ejection in the final concentration indicated. Two-photon laser uncaging and optogenetic experiments were performed in the presence of 2 µM TTX and 200 µM 4-AP. The ROCK2 inhibitor SR3850 was obtained from CHDI (CHDI-00484785).

## Statistical analysis

Data was graphically presented using non-parametric box and whisker plots. In these plots, the center line is the median, the edges of the box mark the interquartiles of the distribution and the lines extend to the limiting values of the sample distribution; outliers (defined as values further away from the median than 1.5XInterquartile range) are marked as asterisks. Statistical significance was determined using non-parametric tests running on GraphPad Prism Version 5.0 (GraphPad Software).


# Acknowledgments

We thank Dr. Vivian Hernandez for initial characterization of the Kv4.2 shRNA. Sasha Ulrich for technical assistance. This work was supported by CHDI (DJS), JPB (DJS and MK), and NIH (NS 069777 to CSC and NS 34696 to DJS).


# Additional information


### Funding

| Funder | Grant reference number | Author |
|---|---|---|
| CHDI Foundation | | D James Surmeier |
| JPB Foundation | | D James Surmeier |
| National Institutes of Health | NS069777 | C Savio Chan |
| National Institutes of Health | NS34696 | D James Surmeier |

The funders had no role in study design, data collection and interpretation, or the decision to submit the work for publication.


### Author contributions

Luis Carrillo-Reid, Conceptualization, Data curation, Formal analysis, Investigation, Methodology, Responsible for the design, execution and analysis of experiments pertaining to the dendritic excitability deficit in BACHD mice and its modulation; Michelle Day, Conceptualization, Data curation, Formal analysis, Investigation, Methodology, Responsible for the design, execution and analysis of the dendritic excitability experiments in Q175 mice and the ZFP experiments; Zhong Xie, Data curation, Formal analysis, Investigation, Methodology, Responsible for the co-immunoprecipitation experiments; Alexandria E Melendez, Data curation, Formal analysis, Investigation, Methodology, Conducted and helped design the dendritic excitability experiments; Jyothisri Kondapalli, Data curation, Formal analysis, Investigation, Methodology, Responsible for the testing of Kv4.2 shRNAs and for the development and delivery of the ZFP; Joshua L Plotkin, Data curation, Investigation, Methodology, Assisted in the design of the experiments and in the testing of the ROCK inhibitors; David L Wokosin, Conceptualization, Formal analysis, Investigation, Methodology, Assisted in the design and analysis of the dendritic excitability measurements after ZFP delivery; Yu Chen, Data curation, Formal analysis, Investigation, Methodology, Conducted the qPCR experiments and stereotaxic injections of AAV carrying ZFP; Geraldine J Kress, Data curation, Formal analysis, Investigation, Participated in the PTEN experiments; Michael Kaplitt, Resources, Methodology, Designed and tested the shPTEN construct; Ema Ilijic, Data curation, Formal analysis, Methodology, Conducted Kv4.2 dendritic colocalization analysis; Jaime N Guzman, Conceptualization, Data curation, Investigation, Methodology, Participated in the design and conduct of the initial dendritic excitability experiments; C Savio Chan, Conceptualization, Data curation, Formal analysis, Investigation, Methodology, Responsible for design and cloning of the Kv4.2 shRNA and the design and analysis of qPCR

experiments; D James Surmeier, Conceptualization, Formal analysis, Supervision, Funding acquisition, Visualization, Writing—original draft, Project administration, Writing—review and editing, Responsible for the overall direction of the project, preparation of figures and writing of the manuscript

### Author ORCIDs
Luis Carrillo-Reid https://orcid.org/0000-0002-0511-2533
Michelle Day https://orcid.org/0000-0002-9473-0527
Zhong Xie https://orcid.org/0000-0002-8348-4455
Joshua L Plotkin https://orcid.org/0000-0001-6232-7613
Yu Chen https://orcid.org/0000-0003-0896-1335
Geraldine J Kress https://orcid.org/0000-0001-9442-8734
D James Surmeier https://orcid.org/0000-0002-6376-5225

### Ethics
Animal experimentation: All procedures conformed to the guidelines of Northwestern University Animal Care and Use Committee (reference number IS00001185).

### Decision letter and Author response
Decision letter https://doi.org/10.7554/eLife.40818.038
Author response https://doi.org/10.7554/eLife.40818.039

## Additional files

### Supplementary files
- Transparent reporting form

DOI: https://doi.org/10.7554/eLife.40818.036

### Data availability
All data generated or analysed during this study are included in the manuscript and supporting files. Source data files have been provided for all applicable figures.

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
