## [Decision Letter]

Thank you for submitting your article "Mutant huntingtin enhances activation of dendritic Kv4 K^+^ channels in striatal spiny projection neurons" for consideration by *eLife*. Your article has been reviewed by three peer reviewers, one of whom is a member of our Board of Reviewing Editors, and the evaluation has been overseen by Gary Westbrook as the Senior Editor. The following individuals involved in review of your submission have agreed to reveal their identity: Anatol C Kreitzer (Reviewer #2); James Trimmer (Reviewer #3). The reviewers have discussed the reviews with one another and the Reviewing Editor has drafted this decision to help you prepare a revised submission.

Summary:

This is a rigorous and comprehensive study implicating malfunction of Kv4 channels in the distal dendrites of striatal neurons in two mouse models of Huntington's disease (HD). The study shows that excitability in the distal, but not proximal, dendrites of indirect pathway spiny neurons (iSPNs) is sharply regulated by Kv4 currents, and that in slices from the mutant lines there is a reduction in function that can be explained by an increase in these currents. The authors do a thorough study of the mechanism, using down-regulation of Kv4 through shRNA, antibodies to KChIP2, and modulation of signaling pathways known to be impaired in HD. They further provide evidence that signaling through the TrkBR rescues the deficits in dendritic excitability, and is associated with a decrease in association of KChIP2 with Kv4.2 (and presumably with Cav3.2, although this was not assayed). The results are consistent with the model that in HD models, the distal dendrites of iSPNs have reduced excitability that can be rescued with reduction of Kv4 activation, inhibiting ROCK, or providing BDNF. They go a step further, showing that the effects of the mutant huntingtin is cell autonomous by expressing a zinc finger protein that reduces the mHtt mRNA only in iSPNs. In these cells, dendritic excitability was maintained in a normal state, supporting the cell autonomous hypothesis. Knocking down expression of mutant huntingtin protein in striatum restored dendritic excitability and LTP in spiny projection neurons, providing evidence supporting a causal link between the presence of the mutant protein and altered dendritic signaling. The authors point out that their zinc finger protein also has translational potential.

Essential revisions:

1) The LTP data presented in Figure 7 is a bit of a disconnect with the other presented data. An implicit assumption that could be made by readers is that the mechanisms described in Figures 1-6 (related to Kv4 channels and dendritic excitability) somehow give rise to the loss of LTP. While there may be some relationship, it is not explicitly tested in Figure 7, where the authors dialyze iSPNs with Cesium to block K-channels. The results in Figure 7 seem like a non-sequitur, showing that the ZFP can also restore LTP (although almost certainly through a different mechanism in those experiments). The authors should make it clear in the manuscript, ideally in both the Results section and Discussion section, that the rescue of LTP by the ZFP is not through the same mechanism described in the rest of the paper (regulation of Kv4 channels). Alternatively, the last figure could simply be deleted.

2) The evidence supporting the overall molecular mechanism as presented in Figure 4A and 4F, namely that mutant huntingtin protein and BDNF signaling impact dendritic excitability through effects on the association of Kv4.2/KChIP2/Cav3.2. The authors should perform multilabel immunohistochemistry experiments to determine whether the expression levels of these proteins is altered in the distal dendrites of spiny projection neurons (i.e., in the striatal neuropil) of HD mice versus WT. These data would provide a huge increase in spatial resolution above the samples of homogenized whole striatum that were used for the RT-PCR and immunoblot experiments, and could be performed on samples from the same mice used throughout this study that selectively express GFP in iSPNs, so proximal versus distal dendrites of GFP-positive iSPNs can be distinguished. All of the antibodies used here work well for immunofluorescence immunohistochemistry in mouse brain. The analyses of mRNA and protein levels samples prepared from striatum do not provide spatial resolution as to expression levels in the site under study (the distal dendrites of spiny projection neurons), and given the established role of KChIPs in trafficking, one could imagine a scenario in which the overall expression levels of Kv4.2 are unchanged, but its levels in distal dendrites are substantially altered. This would also address whether the ratio of Kv4.2 and KChIP2 are altered in distal dendrites, and whether there is replacement of KChIP2 with one of the other KChIPs present in these cells and that may not have the same impact on Kv4.2 and/or the Kv4:KChIP:Cav3 complex.

3) The co-IP experiments that further support this model should also be strengthened. Specifically, the authors use non-immune IgG as a negative control. While this is commonly used, it is not rigorous as IgG pulls down nothing, and does not provide compelling evidence that what is pulled down with a target-specific antibody that is effective at IP'ing something is specifically co-IP'd with that target, or would come down with the effective IP of any target protein from the same samples. A much stronger control is to successfully IP related but distinct proteins present in these samples (e.g., other potassium channel subunits such as Kv1 or Kv2 channels and their auxiliary subunits) and show that these IPs have the target but lack the components of what the authors argue is the dedicated and specific association of Kv4.2:KChIP2:Cav3.2.

4) Much of the data regarding physiology and rescue can also be explained by reductions in Cav conductances that can't be detected using mRNA levels. When the authors downregulate/down-modify closely associated K channels/subunits, more Ca will come in. Are there any further data that rule out a primary role of Ca channel pathology? If so, please add these; if not, the authors need to consider and discuss this in the manuscript. The experiments suggested above under 2 and 3 will also help to provide information on this point.

---

## [Author Response]

Essential revisions:1) The LTP data presented in Figure 7 is a bit of a disconnect with the other presented data. An implicit assumption that could be made by readers is that the mechanisms described in Figures 1-6 (related to Kv4 channels and dendritic excitability) somehow give rise to the loss of LTP. While there may be some relationship, it is not explicitly tested in Figure 7, where the authors dialyze iSPNs with Cesium to block K-channels. The results in Figure 7 seem like a non-sequitur, showing that the ZFP can also restore LTP (although almost certainly through a different mechanism in those experiments). The authors should make it clear in the manuscript, ideally in both the Results section and Discussion section, that the rescue of LTP by the ZFP is not through the same mechanism described in the rest of the paper (regulation of Kv4 channels). Alternatively, the last figure could simply be deleted.

The reason for including this data was actually relatively simple. Our working hypothesis was that lowering mHtt would restore TrkBR signaling to iSPNs and in so doing re-establish normalize dendritic excitability. It did so. However, it was possible that lowering mHtt was altering dendritic excitability in other ways. Another dendritic phenomenon that is dependent upon TrkBR signaling is LTP. Sure enough, ZFP-mediated lowering of mHtt also restored LTP, providing added confidence to the assertion that TrkBR signaling had been rescued. In the revision, the logic is more clearly laid out and it is stated that the restoration of LTP is not a consequence of restoration of Kv4.2 function but rather that of TrkBR signaling. Moreover, references to the LTP experiments have been eliminated from the Discussion section.

2) The evidence supporting the overall molecular mechanism as presented in Figure 4A and 4F, namely that mutant huntingtin protein and BDNF signaling impact dendritic excitability through effects on the association of Kv4.2/KChIP2/Cav3.2. The authors should perform multilabel immunohistochemistry experiments to determine whether the expression levels of these proteins is altered in the distal dendrites of spiny projection neurons (i.e., in the striatal neuropil) of HD mice versus WT. These data would provide a huge increase in spatial resolution above the samples of homogenized whole striatum that were used for the RT-PCR and immunoblot experiments, and could be performed on samples from the same mice used throughout this study that selectively express GFP in iSPNs, so proximal versus distal dendrites of GFP-positive iSPNs can be distinguished. All of the antibodies used here work well for immunofluorescence immunohistochemistry in mouse brain. The analyses of mRNA and protein levels samples prepared from striatum do not provide spatial resolution as to expression levels in the site under study (the distal dendrites of spiny projection neurons), and given the established role of KChIPs in trafficking, one could imagine a scenario in which the overall expression levels of Kv4.2 are unchanged, but its levels in distal dendrites are substantially altered. This would also address whether the ratio of Kv4.2 and KChIP2 are altered in distal dendrites, and whether there is replacement of KChIP2 with one of the other KChIPs present in these cells and that may not have the same impact on Kv4.2 and/or the Kv4:KChIP:Cav3 complex.

We agree that a definitive anatomical analysis of iSPN dendrites in Q175 and wildtype mice would have strengthened our conclusions about the mechanisms underlying the change in dendritic excitability. However, to our knowledge, this type of analysis has never been performed in brain tissue from adult mice. Nevertheless, we recruited an expert (Ema Ilijic, added as a co-author) in the immunocytochemical localization of proteins to the project in an attempt to do so. To this end, we used sparse labeling of iSPNs using AAV delivery of a Cre-dependent fluorescent reporter to fill iSPNs in 6 month-old wildtype and Q175 mice. Next, Kv4.2 immunoreactivity was localized using well-characterized antibodies (NeuroMab). Confocal microscopy was then used to acquire images of the striatum. Lastly, z-stacks obtained in this way were deconvolved and then analyzed using ImarisColoc to determine the distribution of Kv4.2 protein in proximal and distal dendrites of iSPNs. This new set of studies found no difference in the dendritic distribution of Kv4.2 protein in wildtype and zQ175 iSPNs (Figure 3—figure supplement 2). In addition, the Discussion section has been edited to make the reader aware of the possibility that we cannot exclude the possibility that there are other combinations of the channel complex that were differentially distributed.

3) The co-IP experiments that further support this model should also be strengthened. Specifically, the authors use non-immune IgG as a negative control. While this is commonly used, it is not rigorous as IgG pulls down nothing, and does not provide compelling evidence that what is pulled down with a target-specific antibody that is effective at IP'ing something is specifically co-IP'd with that target, or would come down with the effective IP of any target protein from the same samples. A much stronger control is to successfully IP related but distinct proteins present in these samples (e.g., other potassium channel subunits such as Kv1 or Kv2 channels and their auxiliary subunits) and show that these IPs have the target but lack the components of what the authors argue is the dedicated and specific association of Kv4.2:KChIP2:Cav3.2.

As suggested, a co-IP was performed using a Kv1.2 antibody to determine the specificity of the pull-down. Kv1.2 protein is abundant in the striatum (Figure 4—figure supplement 1B) and was pulled down by the Kv1.2 antibody (Figure 4—figure supplement 1B). However, the Kv1.2 antibody did not pull-down KChIP2 (Figure 4—figure supplement 1B). These results support the proposition that the association between Kv4.2 and KChIP2 was specific.

4) Much of the data regarding physiology and rescue can also be explained by reductions in Cav conductances that can't be detected using mRNA levels. When the authors downregulate/down-modify closely associated K channels/subunits, more Ca will come in. Are there any further data that rule out a primary role of Ca channel pathology? If so, please add these; if not, the authors need to consider and discuss this in the manuscript. The experiments suggested above under 2 and 3 will also help to provide information on this point.

The ability of the Kv4 channel inhibitor 4-AP to normalize EPSPs evoked by 2P uncaging of glutamate on distal dendrites of Q175 iSPNs argues against a primary role for Cav3 channels in the shift in dendritic excitability. Nevertheless, in the Discussion section of the revision, the possibility that altered function of distal Cav3 channels contributed to the observations is acknowledged.